# Studying the Interplay Between the Actor and Critic Representations in Reinforcement Learning

**Samuel Garcin**[*]
University of Edinburgh

**Trevor McInroe**[*]
University of Edinburgh

**Pablo Samuel Castro**
Google DeepMind, Mila

**Prakash Panangaden**
McGill, Mila

**Christopher G. Lucas**
University of Edinburgh

**David Abel**
Google DeepMind, University of Edinburgh

**Stefano V. Albrecht**
University of Edinburgh

## Abstract

Extracting relevant information from a stream of high-dimensional observations is a central challenge for deep reinforcement learning agents. Actor-critic algorithms add further complexity to this challenge, as it is often unclear whether the same information will be relevant to both the actor and the critic. To this end, we here explore the principles that underlie effective representations for the actor and for the critic in on-policy algorithms. We focus our study on understanding whether the actor and critic will benefit from separate, rather than shared, representations. Our primary finding is that when separated, the representations for the actor and critic systematically specialise in extracting different types of information from the environment—the actor's representation tends to focus on action-relevant information, while the critic's representation specialises in encoding value and dynamics information. We conduct a rigourous empirical study to understand how different representation learning approaches affect the actor and critic's specialisations and their downstream performance, in terms of sample efficiency and generation capabilities. Finally, we discover that a separated critic plays an important role in exploration and data collection during training. Our code, trained models and data are accessible at `https://github.com/francelico/deac-rep`.

## 1 Introduction

In recent years, auxiliary representation learning objectives have become increasingly prominent in deep reinforcement learning (RL) agents (Yarats et al., 2021a; Dunion et al., 2023a). These objectives facilitate extracting relevant features from high dimensional observations, and can help improve the sample efficiency and generalisation capabilities of both value-based (Anand et al., 2019; Schwarzer et al., 2021) and actor-critic methods (Yarats et al., 2021b; Zhang et al., 2021; McInroe et al., 2023). However, knowing whether a particular representation learning objective will work and understanding *why* it works is often difficult due to the interplay between the different components of modern RL algorithms.

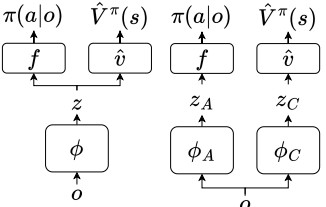

Figure 1: Models with shared (left) and decoupled representations (right).

Online actor-critic algorithms like PPO (Schulman et al., 2017) jointly optimise policy improvement and value estimation objectives. When parametrised by deep neural networks, the actor (in charge of improving the policy) and the critic (in charge of estimating the value of the current policy) often share the same learned representation $\phi$, which maps observations to latent features $z$.

---

[*]Equal contribution. Correspondence to {`s.garcin,t.mcinroe`}`@ed.ac.uk`

Cobbe et al. (2021) and Raileanu & Fergus (2021) report that fully separating the actor and critic networks, i.e. *decoupling* the two (Figure 1, right), improves sample efficiency and generalisation over a shared architecture (Figure 1, left). We hypothesise that decoupling is effective because it encourages information specialisation in $\phi_A$ and $\phi_C$, which in turn improves performance. To test our hypothesis, we introduce metrics to quantify specialisation, and we conduct an extensive empirical study of three on-policy actor-critic algorithms (Schulman et al., 2017; Cobbe et al., 2021; Moon et al., 2022) across discrete and continuous control benchmarks, and under various representation learning approaches (Raileanu & Fergus, 2021; Moon et al., 2022; Raileanu et al., 2021; Castro et al., 2021) applied to the actor, to the critic, or to both. We supplement this empirical study by a theoretical characterisation of the information extracted by the actor and critic's respective optimal representations.

Table 1: Once decoupled, the actor and critic representations $\phi_A$ and $\phi_C$ specialise in capturing different information from the environment. Reported values correspond to a PPO agent trained in Procgen (Cobbe et al., 2020). See §2 and §3 for formal definitions of the quantities quoted.

| If ... is high, | it is possible to ... | % change from using a shared representation | |
| --- | --- | --- | --- |
| | | $\phi_A$ | $\phi_C$ |
| $\mathrm{I}(Z;L)$ | overfit to training levels (environment instances). | -20% | +35% |
| $\mathrm{I}(Z;V)$ | use $z$ to predict state values. | +37% | +41% |
| $\mathrm{I}((Z,Z');A)$ | use $z$ and $z'$ obtained from consecutive timesteps $t, t'$ to identify the action taken at timestep $t$. | +23% | -48% |
| $\mathrm{I}(Z;Z')$ | differentiate between latent pairs obtained from consecutive and non-consecutive timesteps. | -96% | +324% |

Our main findings are summarised below.

- Decoupled actor and critic representations extract different information about the environment. This information specialisation, described and quantified in Table 1, systematically occurs in the on-policy algorithms and benchmarks covered by our study, and is consistent with the actor's and critic's respective optimal representations.

- The actor benefits from representation learning approaches that prioritise extracting level-invariant information over level-specific information. This bias for level-invariant information matters more than what specific information quantity is targeted by the representation learning objective. Nevertheless, approaches antithetical to the actor's inherent information specialisation tend to perform poorly.

- Through its role as a baseline in the actor's objective, a decoupled critic will tend to bias policy updates to facilitate the optimisation of its own learning objective. The critic, therefore, plays an important role in exploration and data collection during training. Thus, we find that care must be taken when selecting a representation learning objective for the critic: certain objectives improve the critic's value predictions but may prevent convergence to the optimal policy because the objective induces significant bias.

## 2 BACKGROUND

**RL framework.** We follow the framework established by Kirk et al. (2023), which, given a fixed timestep budget, lets us quantify the agent's performance in its original training environment (i.e. its sample efficiency) and its performance in held-out instances (i.e. its generalisation capabilities). We consider the episodic setting, and model the environment as a Contextual-MDP (CMDP) $\mathcal{M} \triangleq (\mathbb{S}, \mathbb{A}, \mathbb{O}, \mathcal{T}, \Omega, R, \mathbb{C}, P(\mathrm{c}), \mathcal{P}_0, \gamma)$ with state, action and observation spaces $\mathbb{S}, \mathbb{A}$ and $\mathbb{O}$ and discount factor $\gamma$. In a CMDP, the reward $R : \mathbb{S} \times \mathbb{C} \times \mathbb{A} \to \mathbb{R}$, and observation functions $\Omega : \mathbb{S} \times \mathbb{C} \to \mathbb{O}$ as well as the transition $\mathcal{T} : \mathbb{S} \times \mathbb{C} \times \mathbb{A} \to \mathscr{P}(\mathbb{S})$, and initial state $\mathcal{P}_0 : \mathbb{C} \to \mathscr{P}(\mathbb{S})$ kernels can change with the *context* $c \in \mathbb{C}$, with $c \sim P(\mathrm{c})$ at the start of each episode. The CMDP is

therefore conceptually equivalent to an MDP with state space $\mathbb{X} : \mathbb{S} \times \mathbb{C}$. Each context $c$ maps one-to-one to a particular environment instance, or *level*, and thus represents the component of the state $x$ that cannot change during the episode. The agent's policy $\pi : \mathbb{O} \rightarrow \mathscr{P}(\mathbb{A})$ maps observations to action distributions and induce a value function $V^\pi : \mathbb{X} \rightarrow \mathbb{R}$ mapping states to expected future returns $V^\pi(x) = \mathbb{E}_\pi[\sum_t^T \gamma^t r_t]$, where $\{r_t\}_{0:T}$ are possible sequences of rewards obtainable when following policy $\pi$ from $x$ and until the episode terminates. We define the optimal policy $\pi^*$ as the policy maximising expected returns $\mathbb{E}_{c \sim P(c), x_0 \sim \mathcal{P}_0(c)}[V^\pi(x_0)]$. During training, we assume access to a limited set of training levels $L \sim P(c)$. We evaluate sample efficiency by measuring returns over $L$ and generalisation by evaluating on an held-out set $L_{\text{test}} \sim P(c)$.

**Actor-critic architectures.** On-policy actor-critic models consist of a policy network $\pi_{\boldsymbol{\theta}_A}$ and a value network $\hat{V}_{\boldsymbol{\theta}_C}$, with *actor* parameters $\boldsymbol{\theta}_A$ and *critic* parameters $\boldsymbol{\theta}_C$ (we use $\cdot_A/\cdot_C$ when referring to the actor/critic in this work). When learning from high dimensional observations, such as pixels, a representation $\phi : \mathbb{O} \rightarrow \mathbb{Z}$ maps observations to latent features $z \in \mathbb{Z}$. When coupled, the policy and value networks share a representation and split into actor and critic heads $f$ and $\hat{v}$. That is, we have $\pi_{\boldsymbol{\theta}_A} \triangleq f_{\boldsymbol{\omega}} \circ \phi_{\boldsymbol{\eta}}$ and $\hat{V}_{\boldsymbol{\theta}_C} \triangleq \hat{v}_{\boldsymbol{\xi}} \circ \phi_{\boldsymbol{\eta}}$, with $\boldsymbol{\theta}_A \triangleq (\boldsymbol{\omega}, \boldsymbol{\eta})$ and $\boldsymbol{\theta}_C \triangleq (\boldsymbol{\xi}, \boldsymbol{\eta})$. When decoupled, two representation functions $\phi_A$, $\phi_C$ with parameters $(\boldsymbol{\eta}_A, \boldsymbol{\eta}_C)$ are learned.

**PPO and PPG.** In this work, we investigate the representation properties of PPO (Schulman et al., 2017) and Phasic Policy Gradient (PPG) (Cobbe et al., 2021), two actor-critic algorithms that have been reported to benefit from improved sample efficiency and transfer upon decoupling (Raileanu & Fergus, 2021; Cobbe et al., 2021). In PPO, the actor maximises

$$J_\pi(\boldsymbol{\theta}_A) = \mathbb{E}_B\left[\min(\frac{\pi_{\boldsymbol{\theta}_A}(a_t|o_t)}{\pi_{\boldsymbol{\theta}_{A\,old}}(a_t|o_t)}\hat{A}_t, \text{clip}(\frac{\pi_{\boldsymbol{\theta}_A}(a_t|o_t)}{\pi_{\boldsymbol{\theta}_{A\,old}}(a_t|o_t)}, 1-\epsilon, 1+\epsilon)\hat{A}_t) + \beta_H\mathbf{H}(\pi_{\boldsymbol{\theta}_A}(a_t|o_t))\right], \tag{1}$$

where $\boldsymbol{\theta}_{A\,old}$ are the actor weights before starting a round of policy updates, $B$ is a batch of trajectories collected with $\pi_{\boldsymbol{\theta}_{A\,old}}$, $\hat{A}_t$ is an estimator for the advantage function at timestep $t$, $\mathbf{H}(\cdot)$ denotes the entropy and $\epsilon$ and $\beta_H$ are hyperparameters controlling clipping and the entropy bonus. The critic minimises

$$\ell_V(\boldsymbol{\theta}_C) = \frac{1}{|B|}\sum_{o_t \in B}(\hat{V}_{\boldsymbol{\theta}_C}(o_t) - \hat{V}_t)^2, \tag{2}$$

where $\hat{V}_t$ are value targets. Both $\hat{A}$ and $\hat{V}$ are computed using GAE (Schulman et al., 2016). PPG performs an auxiliary phase after conducting PPO updates over $N_\pi$ policy phases. To prevent overfitting, the auxiliary phase fine-tunes the critic and distills value information into the representation from much larger trajectory batches $B_{aux} = \bigcup_{i \in 1,...,N_\pi} B_i$, using the loss $\ell_{\text{joint}} = \ell_V + \ell_{aux}$, with

$$\ell_{aux}(\boldsymbol{\theta}_A) = \frac{1}{|B_{aux}|}\sum_{(a_t,o_t) \in B_{aux}}(\hat{V}^{aux}_{\boldsymbol{\theta}_A}(o_t) - \hat{V}_t)^2 + \beta_c D_{\text{KL}}(\pi_{\boldsymbol{\theta}_{A\,old}}(a_t|o_t)\|\pi_{\boldsymbol{\theta}_A}(a_t|o_t)), \tag{3}$$

where $\beta_c$ controls the distortion of the policy. When decoupled, $\hat{V}^{aux} \triangleq v^{aux} \circ \phi_A$ distills value information into representation parameters $\boldsymbol{\eta}_A$ through an additional head $v^{aux}$. When coupled, $v^{aux} \equiv \hat{v}$, and a stop-gradient operation on $\ell_V$ ensures $\boldsymbol{\eta}$ is updated by the critic during the auxiliary phase only.

**Mutual information.** We study the information embedded in features $z$ outputted by $\phi$. To do so, we propose metrics based on the mutual information $\mathrm{I}(X; Y)$, measuring the information shared between sets of random variables $X$ and $Y$, defined as

$$\mathrm{I}(X; Y) = \mathbf{H}(X) + \mathbf{H}(Y) - \mathbf{H}(X, Y) = \sum_{\mathbb{X}}\sum_{\mathbb{Y}} p(x, y) \log \frac{p(x, y)}{p(x)p(y)}, \tag{4}$$

where integrals replace sums for continuous quantities. $\mathrm{I}(X; Y)$ is symmetric, and quantifies how much information about $Y$ is obtained by observing $X$, and vice versa. Similarly, the conditional mutual information $\mathrm{I}(X; Y|Z)$ measures the information shared between $X$ and $Y$ that does not depend on $Z$. We measure mutual information using the k-nearest neighbors entropy estimator proposed by Kraskov et al. (2004) and extended to pairings of continuous and discrete variables by Ross (2014). We briefly introduce notation for random variables used in following sections. $L \sim P(c)$ denotes the set of training levels drawn from the CMDP context distribution. $A$, $R$, $O$,

$O'$, $X$ and $X'$ are sets constructed from $n$ transitions $(a_t, r_t, o_t, o_{t+1}, x_t, x_{t+1})$ uniformly sampled from a batch of trajectories collected in $L$ using policy $\pi$. $Z$ and $Z'$ are latents features, with $z = \phi(o), z' = \phi(o')$. We construct $V$ using the rewards obtained from $t$ until episode termination, with $v_t = \sum_{\bar{t}=t}^{T} \gamma^{\bar{t}-t} r_{\bar{t}}$.

# 3 CATEGORISING AND QUANTIFYING THE INFORMATION EXTRACTED BY LEARNED REPRESENTATIONS

To conduct our analysis of the respective functions of the actor and critic representations, we analyse the information being extracted from observations at agent convergence. We propose four mutual information metrics to measure information extracted about the identity of the current training level, the value function, and the inverse and forward dynamics of the environment. Each metric relates to quantities relevant to the actor and critic's respective learning objectives, and to the agent's generalisation performance. They are introduced in turn below.

**Overfitting.** Our first metric, $I(Z; L)$, quantifies overfitting of the actor and critic representations to the set of training levels, as it measures how easy it is to infer the identity of the current level from $Z$. We follow a similar reasoning as Garcin et al. (2024) to derive an upper bound for the generalisation error that is proportional to $I(Z_A; L)$.[1]

**Theorem 3.1.** *The difference in returns achieved in train levels and under the full distribution, or generalisation error, has an upper bound that depends on $I(Z_A; L)$, with*

$$\mathbb{E}_{c \sim \mathcal{U}(L), x_0 \sim \mathcal{P}_0(c)}[V^\pi(x_0)] - \mathbb{E}_{c \sim P(c), x_0 \sim \mathcal{P}_0(c)}[V^\pi(x_0)] \leq \sqrt{\frac{2D^2}{|L|} \times I(Z_A; L)}, \quad (5)$$

*where $c \sim \mathcal{U}(L)$ indicates $c$ is sampled uniformly over levels in $L$, $D$ is a constant such that $|V^\pi(x)| \leq D/2, \forall x, \pi$ and $Z_A$ is the output space of the actor's learned representation.*

**Value information.** The second metric quantifies $I(Z; V)$, the mutual information between $Z$ and state values. While a high $I(Z_C; V)$ facilitates the minimisation of $\ell_V$ (Equation (2)), we wish to understand whether increasing $I(Z_A; V)$ is always desirable. In fact, we find an apparent contradiction on this matter in prior work: Cobbe et al. (2021) and Wang et al. (2023) report that value distillation into the actor's representation (which implies a high $I(Z_A; V)$) improves sample efficiency and generalisation of coupled and decoupled PPG agents, whereas Raileanu & Fergus (2021) and Garcin et al. (2024) report a positive correlation between generalisation and a high $\ell_V$ for coupled PPO agents (which implies a low $I(Z_A; V)$).

**Dynamics in the latent space.** The remaining two metrics investigate the transition dynamics $\mathcal{T}_z : \mathbb{Z} \times \mathbb{A} \to \mathscr{P}(\mathbb{Z})$ within the latent state space $\mathbb{Z}$ spanned by the representation. We will see in later sections that the *reduced MDP* $(\mathbb{Z}, \mathbb{A}, \mathcal{T}_z, R_z, \gamma)$ spanned by the actor or critic's representation tend to have distinct $\mathcal{T}_z$, which often markedly differ from the transition dynamics $\mathcal{T}$ in the original environment. $I(Z; Z')$ measures how easy it is to differentiate a latent pair $(\phi(o), \phi(o'))$ obtained from consecutive observations from a latent pair obtained from non-consecutive observations. $I((Z, Z'); A)$ quantifies how easy it is to predict the action taken during transition $\langle o, a, o' \rangle$ given the pair $(\phi(o), \phi(o'))$. In Theorem 3.2, we establish that $\mathcal{T}_z$ maintains the *Markov property* of the original MDP when both of these metrics attain their theoretical maximum.[2]

**Theorem 3.2.** *if $\mathcal{T} : \mathbb{X} \times \mathbb{A} \to \mathscr{P}(\mathbb{X})$ satisfies the Markov property, and we have $I((X, X'); A) = I((Z, Z'); A)$ and $I(X; X') = I(Z; Z')$ for any $X, X', A, Z, Z'$ collected using policy $\pi$, then $\mathcal{T}_z : \mathbb{Z} \times \mathbb{A} \to \mathscr{P}(\mathbb{Z})$ satisfies the Markov property when following $\pi$. $\mathcal{T}_z$ always satisfies the Markov property if the above conditions hold for any $\pi$.*

Given that $\phi$ only induces $\mathcal{T}_z$ for the current $\pi$ in the on-policy setting, we make the distinction between $\mathcal{T}_z$ being Markov when following $\pi$ and the more general notion of $\mathcal{T}_z$ being Markov when following any policy. Crucially, Theorem 3.2 generalises the equivalence relations obtained by Allen

---

[1]Proofs for the theoretical results presented in this work are provided in Appendix A.

[2]The Markov property is satisfied for a MDP $(\mathbb{Z}, \mathbb{A}, \mathcal{T}_z, R_z, \gamma)$ if and only if $\mathcal{T}_z^{(k)}(z_{t+1}|\{a_{t-i}, z_{t-i}\}_{i=0}^{k}) = \mathcal{T}_z(z_{t+1}|a_t, z_t)$ and $R_z^{(k)}(z_{t+1}|\{a_{t-i}, z_{t-i}\}_{i=0}^{k}) = R_z(z_{t+1}|a_t, z_t), \forall a \in \mathbb{A}, z \in \mathbb{Z}, k \geq 1$.

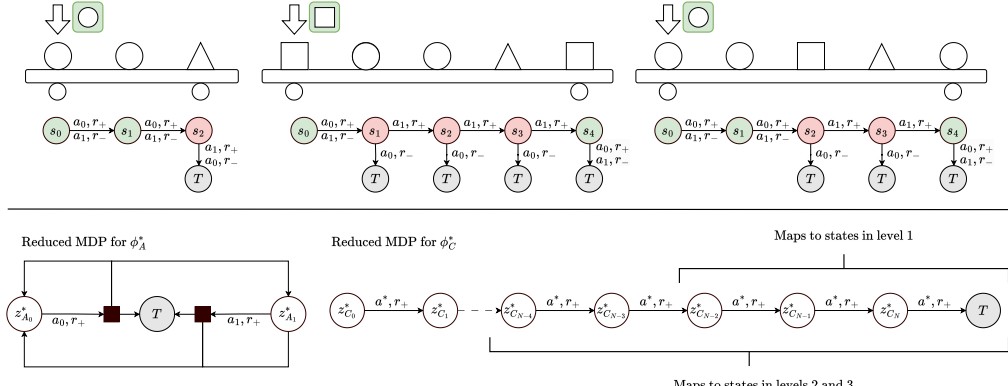

Figure 2: (Top) the initial observations and state spaces of three levels from the assembly line environment in §4. (Bottom) the reduced MDPs spanned by $\phi_A^*$ and $\phi_C^*$.

et al. (2021) to continuous metrics. As such, $\mathrm{I}((Z, Z'); A)$ and $\mathrm{I}(Z; Z')$ quantify how close *any* $\phi$ comes to have $\mathcal{T}_z$ satisfy the Markov property. They remain applicable in settings in which it isn't practical (or even possible) for $\mathcal{T}_z$ to satisfy the Markov property, e.g. when $\phi$ has finite capacity and bottlenecks how much information can be extracted from raw observations, or when observations are not Markov.[3]

## 4 INFORMATION SPECIALISATION IN ACTOR AND CRITIC REPRESENTATIONS

Raileanu & Fergus (2021); Cobbe et al. (2021) have attributed the performance improvements obtained from decoupled architectures to the disappearance of gradient interference between the actor and critic, and to the critic tolerating a higher degree of sample reuse than the actor before overfitting. We propose a different interpretation: given their different learning objectives, the actor's and critic's *optimal representations* (defined below) prioritise different types of information from the environment. While not incompatible with prior interpretations, our claim is stronger. We posit that an optimal (or near-optimal) representation for both the actor and critic will generally be impossible under a shared architecture.

**Definition 4.1.** *Given the model $m \triangleq f_{\boldsymbol{\omega}} \circ \phi$ and associated loss $\ell_m(\boldsymbol{\omega}, \phi)$, an optimal representation $\phi^* : \mathbb{O} \to \mathbb{Z}^*$ satisfies the conditions:*

1. **Optimality conservation.** $\min_{\boldsymbol{\omega}} \ell_m(\boldsymbol{\omega}, \phi^*) = \min_{\boldsymbol{\omega}, \phi} \ell_m(\boldsymbol{\omega}, \phi)$

2. **Maximal compression.** $\phi^* \in \arg\min_{\tilde{\Phi}} |\mathbb{Z}^*|$, *with $\tilde{\Phi}$ the set of all $\phi$ satisfying condition 1.*

We will provide theoretical insights on the respective specialisations and mutual incompatibility of $\phi_A^*$ and $\phi_C^*$, which we further highlight through a motivating example. In our example, depicted in Figure 2, the agent inspects parts for defects on an assembly line. The agent is trained on a set $L$ of levels drawn from $P(c)$. A level is characterised by a particular combination of part specifications, number and ordering, each part having a probability $P^F$ of being defective. At each timestep, the agent observes the part specifications for the current level, which parts are on the assembly line and which part is up for inspection. The agent picks action $a \in \mathbb{A} = \{a_0 = accept, a_1 = reject\}$ and moves to the next part. It receives a reward $R = r_+$ when correctly accepting/rejecting a good/defective part and $R = r_-$ when it makes a mistake, with $r_+ > r_-$. The episode terminates early when the agent accepts a defective part, otherwise it terminates after $N_c$ timesteps, where $N_c$ is the number of parts in level $c \in L$.

---

[3]In contrast, Allen et al. (2021) assume observations are always Markov.

## 4.1 THE ACTOR'S OPTIMAL REPRESENTATION

The combinatorial explosion of possible specifications and part assortments means $\phi_A^*$ should ideally map observations to a *reduced MDP* spanning a much smaller state space that in the original environment. However, $\phi_A^*$ should still provide the information necessary to select the optimal action at each timestep of each level, including those not in the training set.

**Dynamics of the reduced MDP.** Under our definition, the mapping

$$\phi_A^*(o) = \begin{cases} z_{A0}^*, & \text{if } a^* = a_0 \\ z_{A1}^*, & \text{if } a^* = a_1. \end{cases} \tag{6}$$

satisfies the conditions for being an optimal representation, and spans the reduced MDP in Figure 2 (bottom left). This reduced MDP describes the perceived environment dynamics when only observing the latent states in $\mathbb{Z}_A^*$. By construction, $\text{I}((Z_A^*, Z_A^{*\prime}); A)$ is guaranteed to be maximised when following the optimal policy. We note that, even if $\text{I}(X; X')$ is maximised under $\pi^*$ in the original environment, we have $\text{I}(Z_A^*; Z_A^{*\prime}) = 0$ in the reduced MDP. In other words, the current reduced state yields no information about the next reduced state, and vice versa.

**Overfitting to training levels.** In our assembly line example, an *overfit* $\phi_A$ with high $\text{I}(Z_A; L)$ may leverage this information to first identify, and then solve certain levels. This overfit $\phi_A$ may be optimal over $L$, but would be heavily biased to the training set, and fail to generalise to unseen levels that do not satisfy certain spurious correlations. For example: "*if there are three objects on the line then I must be in level 1, and, in level 1, I should reject the triangle*". In contrast, a representation that is *invariant* to individual levels, i.e. with $\text{I}(Z_A; L) = 0$, guarantees zero generalisation error under Theorem 3.1. Nevertheless, achieving $\text{I}(Z_A; L) = 0$ is not achievable under certain conditions, stated in Lemma 4.1.

**Lemma 4.1.** *$I(Z; L) > 0$ if $I(O; L) > 0$ and $\exists z_k, c_j \in Z \times L$ such that $\mu(z_k|c_j) \neq \mu(z_k)$.*

In our example, $\text{I}(O; L) > 0$, since a level can be identified from its observations. We can verify that the second condition holds for $\phi_A^*$ by inspecting the stationary distributions in a particular level $c$ and over all levels,

$$\mu(z) = \begin{cases} \bar{P}^F, & \text{if } z = z_{A0}^* \\ P^F, & \text{if } z = z_{A1}^* \end{cases} \quad \mu(z|c) = \begin{cases} \bar{P}_c^F, & \text{if } z = z_{A0}^* \\ P_c^F, & \text{if } z = z_{A1}^*, \end{cases} \tag{7}$$

where $\bar{P}^F = 1 - P^F$ and $P_c^F$ is the defect probability when in level $c$. We may have $P^F \neq P_c^F$, since individual levels do not all have the same distribution of defective parts. In other words, while it is useful to reduce $\text{I}(Z_A; L)$ to guard against overfitting, the optimal representation $\phi_A^*$ may carry some irreducible information about level identities.

**Value and dynamics distillation can induce overfitting.** we can employ the chain rule of mutual information to decompose the information $\phi$ captures about some arbitrary quantity $Y$ as

$$\text{I}(Z; Y) = \text{I}(Z; Y|L) + \text{I}(Z; L) - \text{I}(Z; L|Y), \tag{8}$$

where $\text{I}(Z; Y|L)$ is the *level-invariant* information encoded about $Y$, and $\text{I}(Z; L) - \text{I}(Z; L|Y)$ is the *level-specific* information being encoded about $Y$. Thus, encouraging $\phi_A$ to capture extraneous information about state values or transition dynamics can cause $\text{I}(Z_A; L)$ to increase and promote overfitting. In Lemma 4.2, we show that increasing $\text{I}(Z_A; V)$ or $\text{I}(Z_A; Z_A')$ (which is not necessary for obtaining an optimal $\phi_A^*$, as discussed above) can cause $\text{I}(Z_A; L)$ to increase as well.

**Lemma 4.2.** *$I(Z; L)$ monotonically increases with a) $I(Z; V) - I(Z; V|L)$ and b) $I(Z; Z') - I(Z; Z'|L)$.*

In fact, $\text{I}(Z; L)$ will increase when $\text{I}(Z; V) > \text{I}(Z; V|L)$ (or when $\text{I}(Z; Z') > \text{I}(Z; Z'|L)$), that is when the information encoded carries a level-specific component. In our example, the higher state values only occur in a subset of levels, since the optimal value for any given state depends on how many parts are left to inspect. State values therefore inherently contain level-specific information, and may induce overfitting if this information is captured by $\phi_A$. This challenges the notion that value distillation would systematically improve the actor's representation.

The key implications of the above are 1) While achieving zero $\text{I}(Z_A; L)$ is not always possible, a high $\text{I}(Z_A; L)$ implies $\phi_A$ is overfit; 2) Increasing $\text{I}(Z_A; Z_A')$ or $\text{I}(Z_A; V)$ is not necessary for obtaining $\phi_A^*$, and will increase $\text{I}(Z_A; L)$ if the information captured is level-specific.

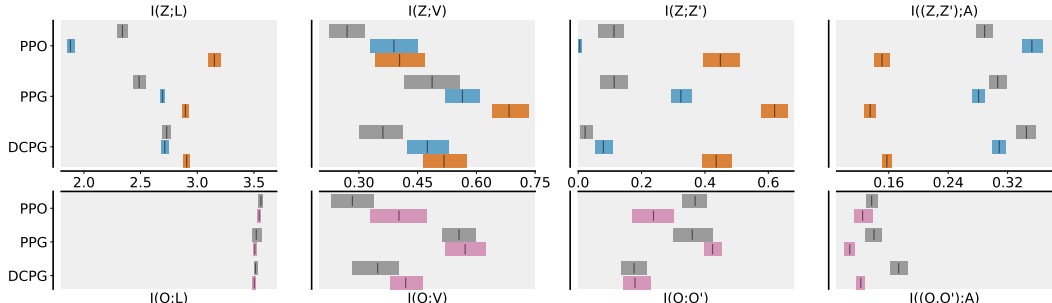

Figure 3: Mean and 95% confidence interval aggregates of $I(Z; \cdot)/I(O; \cdot)$ (top/bottom rows) in Procgen. Gray bars indicate $I(Z; \cdot)/I(O; \cdot)$ for a shared $\phi$. Blue and orange bars indicate $I(Z; \cdot)$ measured for $\phi_A$ and $\phi_C$ when employing a decoupled architecture. Pink bars indicate $I(O; \cdot)$ measured when using a decoupled architecture. X-axes are shared across top and bottom. For all algorithms, decoupling induces specialisation consistent with §4.

## 4.2 THE CRITIC'S OPTIMAL REPRESENTATION

$\phi_C^*$ **has higher** $\mathbf{I}(Z; L)$ **than** $\phi_A^*$**.** The reduced MDP spanned by $\phi_C^*$ is depicted in Figure 2 (bottom right). In order to ensure perfect value prediction, $\phi_C^*$ maps each possible optimal state value to a different element in $\mathbb{Z}_C^*$, and it maximises $I(Z_C^*; V)$ by construction. $I(Z_C^*; Z_C^{*\prime})$ is also high due to the recurrence $V^\pi(x) = \mathbb{E}_{a \sim \pi}[R(x, a) + \gamma \mathbb{E}_{x' \sim P^\pi(x'|x)}[V^\pi(x')]]$. This points to $V^\pi$ being a quantity inherently more level specific than the optimal action for the current state, because $V^\pi$ encodes information pertaining to all possible future states of the current level. We should then expect that, in general, $I(Z_C^*; L) > I(Z_A^*; L)$, implying that decoupling the actor and critic helps with $I(Z_A; L)$ regularisation.

$\phi_C^*$ **is not compatible with** $\pi^*$**.** Paradoxically, while $\phi_C^*$ would necessitate trajectories collected using the optimal policy in order to be learnt, in our example it is not possible to have an optimal policy that only depends on $z_C^*$. The information contained in $z_C^*$ is not sufficient for picking the optimal action in any given timestep, and therefore the best response is to always pick $a_1$ in order to prevent early termination. Therefore, in addition to the information prioritised by $\phi_C^*$ being in general irrelevant to $\pi^*$, employing a shared $\phi$ with a finite capacity may prevent extracting the necessary information to execute the optimal policy.

## 4.3 CONFIRMING SPECIALISATION EMPIRICALLY

We conclude this section by studying the representations learned by PPO (Schulman et al., 2017), PPG (Cobbe et al., 2021) and DCPG (Moon et al., 2022), a close variant of PPG that employs delayed value targets to train the critic and for value distillation. We evaluate all algorithms with and without decoupling their representation. We conduct our experiments in Procgen (Cobbe et al., 2020), a benchmark of 16 games designed to measure generalisation in RL. We report our main observations in below, with extended results and details on our methodology included in Appendix C.2.

Figure 4: Effect of parameter scaling in coupled (blue) and decoupled (orange) PPO. Scores normalized by model performance at 0.6M parameters.

**Specialisation is consistent with** $\phi_A^*$ **and** $\phi_C^*$**.** As no algorithm achieves optimal scores in all games, we now consider the suboptimal representations $\phi_A$ and $\phi_C$ realistically obtainable by the end of training. In Figure 3, we observe clear specialization upon decoupling consistent with the properties we expect for $\phi_A^*$ and $\phi_C^*$. $\phi_C$ has high $I(Z; V)$, $I(Z; Z')$ and $I(Z; L)$, while $\phi_A$ specializes in $I((Z, Z'); A)$ .

**Decoupling is more parameter efficient.** Since decoupled representations fit twice as many parameters, it is fair to wonder whether the performance improvements are mainly caused by the increased model capacity. To test this, we measure performance as we scale model size in a shared and a de-

coupled architecture in Figure 4. Surprisingly, the decoupled model turns out to be *more* parameter efficient, and still outperforms a shared model with four times its parameter count.

**On Markov representations.** According to Theorem 3.2, a representation is considered Markov when both $I((Z, Z'); A)$ and $I(Z; Z')$ are maximized. However, our observations reveal an interesting pattern during decoupling: the actor representation shows an increase in $I((Z_A, Z'_A); A)$ but a decrease in $I(Z_A; Z'_A)$, while the critic representation shows the opposite effect - a decrease in $I((Z_C, Z'_C); A)$ and an increase in $I(Z_C; Z'_C)$. This divergent behavior suggests that neither the actor nor the critic networks inherently benefit from maintaining a Markov representation. This finding aligns with theoretical expectations, as neither $\phi_A$ nor $\phi_C$ need be Markov to be optimal. Furthermore, we found no significant correlation between the sum of these mutual information terms $(I((Z, Z'); A) + I(Z; Z'))$ and agent performance, as shown by comparing Figure 7 and Figure 13.

## 5 REPRESENTATION LEARNING FOR THE ACTOR

In this section, we study how different representation learning objectives affect $\phi_A$ in PPO, PPG and DCPG. We consider advantage (Raileanu & Fergus, 2021) and dynamics (Moon et al., 2022) prediction, data augmentation (Raileanu et al., 2021) and MICo (Castro et al., 2021), an objective explicitly shaping the latent space to embed differences in state values. We study these objectives in Procgen (Figure 5), and in four continuous control environments with video distractors (McInroe & Garcin, 2025)(Figure 11).

**Representation learning impacts information specialisation.** As expected, applying auxiliary tasks alters what information is extracted by the representation. Dynamics prediction generally enhances the specialization of $\phi_A$ by increasing $I((Z, Z'); A)$ while reducing the three other quantities. Conversely, MICo produces the opposite effect - in most cases, it increases $I(Z; Z')$, $I(Z; V)$, and $I(Z; L)$ at the expense of $I((Z, Z'); A)$. The effects of the last two objectives are not as clearcut. Data augmentation produces little change in each quantity, while advantage prediction tends to reduce the measured mutual information, but is inconsistent in the quantities it affects. Performance-wise, data augmentation improves train and test scores for all algorithms; dynamics prediction tends to improve performance for PPG and DCPG; MICo generally decreases performance, and advantage prediction makes no noticeable impact. Based on these findings, we find advisable to not use objectives increasing $I(Z_A; L)$ or playing directly against the specialisation of $\phi_A$.

**On the importance of the batch size and data diversity.** We now turn our attention to an apparent contradiction in the relationship between value distillation and performance. Decoupling PPO, and thus completely forgoing value distillation, leads to improved train and test scores (Figure 13). However, PPG and DCPG perform extensive value distillation (four times as many distillation updates as coupled PPO in our experiments), and achieve significant performance improvements. Crucially, conducting value distillation every $N_\pi$ policy phases ensures the batch size $B_{\text{aux}}$ is $N_\pi$ times larger than the PPO batch size, greatly increasing its data diversity. Wang et al. (2023) report that increasing diversity is a key driver of performance improvement at equal number of gradient updates. We reproduce their experiment in Figure 12, while also tracking the evolution of $I(Z_A; L)$ and $I(Z_A; V)$ as $B_{\text{aux}}$ increases. We find that the increase in $I(Z_A; L)$ in PPG is mainly caused by these additional gradient updates, whereas agent performance and $I(Z_A; V)$ only increase when performing value distillation over large, diverse training batches. Increasing data diversity should promote level-invariant state value information to be distilled into $\phi_A$, and we conclude that it is this level invariant information plays a key role in improving agent performance in PPG and DCPG.

Our experiments further reveal that dynamics prediction yields significant performance improvements, but only when applied to algorithms with large batch sizes (specifically in PPG and DCPG, not in PPO). In comparison, data augmentation, which increases data diversity, demonstrates performance benefits across all three algorithms. These findings suggest an important hypothesis: the actor benefits more from encouraging $\phi_A$ to extract level-invariant information about *any* arbitrary quantity $Y$, rather than from specifically choosing what that quantity $Y$ should be (such as state values or dynamics).

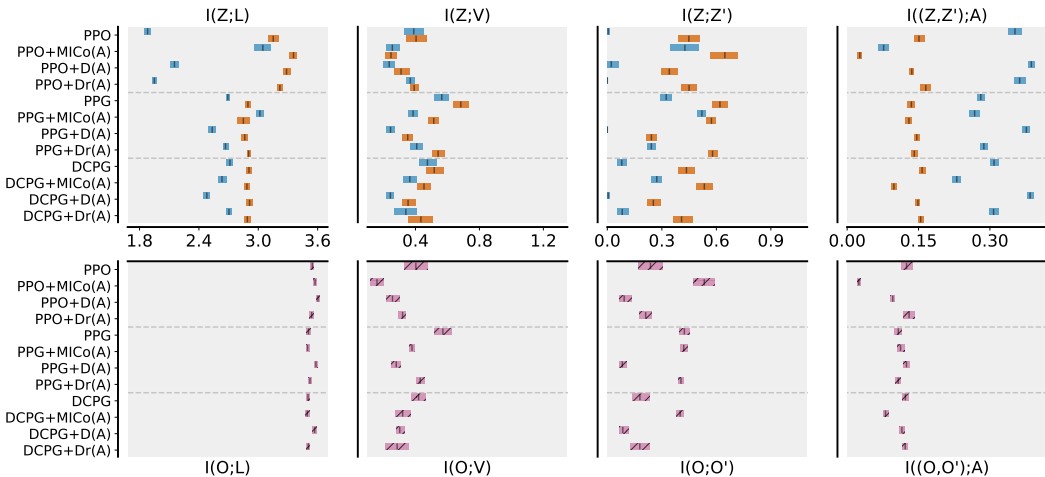

Figure 5: Mean and 95% confidence intervals of $I(Z;\cdot)/I(O;\cdot)$ (top/bottom) for actor (blue) and critic (orange) representations in Procgen. Information measured from agent observations shown in pink. X-axes are shared across top and bottom. Auxiliary tasks shown are MICo, dynamics prediction (D), and data augmentation (Dr) applied to the actor (A).

## 6 THE CRITIC'S OBJECTIVE INFLUENCES DATA COLLECTION

We now consider how the same set of representation learning objectives affect the critic's representation and present our results in Figures 8 and 11. The effect of a given objective on the information extracted by $\phi_C$ is consistent with how they would have affected $\phi_A$ in the previous section. However, we report two surprising findings: a) Without conducting any value distillation, decoupled PPO has a 37% higher $I(Z_A;V)$ than shared PPO (Table 1), and b) the information specialisation of $\phi_C$ incurred by applying an objective on the critic is often observed in $\phi_A$, albeit to a lesser extent. Given that the two representations are decoupled, how can an objective applied to $\phi_C$ affect $\phi_A$?

As we maintain different optimisers for the actor and critic, their only remaining interaction in decoupled PPO is through $J_\pi$ (Equation (1)): $\hat{A}_t$ being computed from the critic's value estimates. Therefore, at least one of the following hypothesis must hold:

1. **Data collection bias.** Through $J^\pi$ updates, the critic biases $\pi$ to collect trajectories containing information relevant to its own learning objective. This information could then leak through $\phi_A$ because more of this information is contained in its input. In this scenario, it is not necessary for $\phi_A$ to become more proficient at extracting critic-relevant information.

2. **Implicit knowledge transfer.** The advantage targets in $J^\pi$ induce information transfer between $\phi_C$ and $\phi_A$ when applying the gradients $\nabla_{\boldsymbol{\theta}_A} J^\pi$. Here, $\phi_A$ becomes proficient at extracting the same information $\phi_C$ extracts.

The first hypothesis broadly holds in our experiments: in most cases, applying MICo to the critic increases $I(O;V)$ and $I(O;O')$, and applying dynamics prediction increases $I((O,O');A)$[4]. Furthermore, $I(O;V)$ increases when PPO is decoupled (Figure 3). Without the critic's influence, there would be no direct incentive for the actor to collect data that contains value information, since no value distillation is taking place.

To test the second hypothesis, we measure the *compression efficiency*, applicable whenever $I(O;\cdot) > 0$, and defined as

$$C(Z;\cdot) = \min\left(\frac{I(Z;\cdot)}{I(O;\cdot)}, 1\right). \tag{9}$$

---

[4]In contrast, I(O;L) does not vary significantly, given that the policy does not control which level is played in an episode.

For example, $C(Z_A; V)$ measures the fraction of available information in $I(O; V)$ that is extracted by $\phi_A$.[5] In Tables 2 and 3, we report that $C(Z_A; V)$ does not signficantly change between PPO[sh] and decoupled PPO, or when MICo is applied to the critic in decoupled PPO. This result appears to disprove the second hypothesis, at least in PPO. We cannot formally confirm whether implicit knowledge transfer occurs for PPG and DCPG, as explicit knowledge transfer already occurs through value distillation.

Finally, we highlight that this data collection bias generally leads to worse performance (Figure 13). Interestingly, employing different representation learning objectives for the actor and the critic results in surprising interactions. Consider PPO in the Procgen setup: when applied separately, Adv(A) and MICo(C) have respectively a neutral and adverse effect on performance, the latter being caused by the data collection bias caused by the critic. When applied together, they exhibit a sharp performance gain, and bring $I(O; V)$ down (Figure 10). This suggests that certain actor representation objectives could improve performance by cancelling the bias in data collection induced by the critic.

## 7 RELATED WORK

**Representation learning in RL.** Representation learning objectives have been used in RL for a variety of reasons such as sample efficiency (Jaderberg et al., 2017; Gelada et al., 2019; Laskin et al., 2020a; Lee et al., 2020; Laskin et al., 2020b), planning (Sekar et al., 2020; McInroe et al., 2024), disentanglement (Dunion et al., 2023b), and generalisation (Higgins et al., 2017; Li et al., 2021). Some works focus on designing metrics motivated by theoretical properties such as bisimulation metrics, pseudometrics, decompositions of MDP components, or successor features (Ferns et al., 2004; Mahadevan & Maggioni, 2007; Dayan, 1993; Castro, 2020; Agarwal et al., 2021; Castro et al., 2021; 2023).

**Analysing representations in RL.** Despite the large body of research into representation learning objectives in RL, relatively little work has gone into understanding the learned representations themselves (Wang et al., 2024). Several works use linear probing to determine how well learned representations relate to environment or agent properties (Racah & Pal, 2019; Guo et al., 2018; Anand et al., 2019; Zhang et al., 2024). Other works analyse the learned representation functions via saliency maps which help visualise where an agent is "paying attention" (Rosynski et al., 2020; Atrey et al., 2020; Dunion et al., 2023a).

## 8 CONCLUSION

In this work, we conducted an in-depth analysis of the representations learned by actor and critic networks in on-policy deep reinforcement learning. Our key findings revealed that when decoupled, actor and critic representations specialise in extracting different types of information from the environment. We found that the actor benefits from representation learning objectives that promote extracting level-invariant information. Finally, we discovered that the critic influences policy updates to collect data that is informative for its own learning objective.

We identify three important research directions to be tackled in future work: 1) Broadening this study's scope to cover different network architectures, representation learning objectives, or RL algorithms (such as off-policy RL (Haarnoja et al., 2018)); 2) Employing these findings to design new representation learning objectives for the actor that target level-invariant information; 3) Leveraging the critic's influence on data collection to devise new exploration strategies in online RL.

## REPRODUCIBILITY STATEMENT

Reproducibility can be challenging without access to the data generated during experiments. To assist with this, we will make all of our experimental data, including model checkpoints, logged

---

[5]By the data processing inequality we must have $I(O; \cdot) \geq I(Z; \cdot)$, and $C(Z; \cdot)$ cannot be larger than 1. We enforce this upper bound as our estimator sometimes underestimates $I(O; \cdot)$ for high dimensional observations.

data and the code for reproducing the figures in this paper openly available at `https://github.com/francelico/deac-rep`.

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

## A  THEORETICAL RESULTS

**Theorem 3.1.** *The difference in returns achieved in train levels and under the full distribution, or generalisation error, has an upper bound that depends on $I(Z_A; L)$, with*

$$\mathbb{E}_{c\sim\mathcal{U}(L),x_0\sim\mathcal{P}_0(c)}[V^\pi(x_0)] - \mathbb{E}_{c\sim P(c),x_0\sim\mathcal{P}_0(c)}[V^\pi(x_0)] \leq \sqrt{\frac{2D^2}{|L|} \times I(Z_A; L)}, \qquad (5)$$

*where $c \sim \mathcal{U}(L)$ indicates $c$ is sampled uniformly over levels in $L$, $D$ is a constant such that $|V^\pi(x)| \leq D/2, \forall x, \pi$ and $Z_A$ is the output space of the actor's learned representation.*

*Proof.* This result directly follows from a result obtained by Bertrán et al. (2020) and reproduced below.

**Theorem A.1.** *For any CMDP such that $|V^\pi(x)| \leq D/2, \forall x, \pi$, with $D$ being a constant, then for any set of training levels $L$, and policy $\pi$*

$$\mathbb{E}_{c\sim\mathcal{U}(L),x_0\sim\mathcal{P}_0(c)}[V^\pi(x_0)] - \mathbb{E}_{c\sim P(c),x_0\sim\mathcal{P}_0(c)}[V^\pi(x_0)] \leq \sqrt{\frac{2D^2}{|L|} \times I(\pi; L)}, \qquad (10)$$

Then, as $\pi \triangleq f \circ \phi_A$, by the data processing inequality we always have $I(\pi; L) \leq I(Z_A; L)$, and therefore,

$$\mathbb{E}_{c \sim \mathcal{U}(L), x_0 \sim \mathcal{P}_0(c)}[V^\pi(x_0)] - \mathbb{E}_{c \sim P(c), x_0 \sim \mathcal{P}_0(c)}[V^\pi(x_0)] \leq \sqrt{\frac{2D^2}{|L|} \times I(\pi; L)}$$

$$\leq \sqrt{\frac{2D^2}{|L|} \times I(Z_A; L)}$$

.

Garcin et al. (2024) follow the same reasoning and obtain an equivalent result, without restating the bound.

$\square$

**Theorem 3.2.** *if* $\mathcal{T} : \mathbb{X} \times \mathbb{A} \to \mathscr{P}(\mathbb{X})$ *satisfies the Markov property, and we have* $I((X, X'); A) = I((Z, Z'); A)$ *and* $I(X; X') = I(Z; Z')$ *for any* $X, X', A, Z, Z'$ *collected using policy* $\pi$, *then* $\mathcal{T}_z : \mathbb{Z} \times \mathbb{A} \to \mathscr{P}(\mathbb{Z})$ *satisfies the Markov property when following* $\pi$. $\mathcal{T}_z$ *always satisfies the Markov property if the above conditions hold for any* $\pi$.

*Proof.* This proof has two part. We first demonstrate that the Inverse Model condition of Theorem A.2 from Allen et al. (2021) (reproduced below) is satisfied if and only if $I((Z, Z'); A) = I((X, X'); A)$. We then show that if $I(Z; Z') = I(X; X')$ then the Density Ratio condition is also satisfied.

**Theorem A.2.** $\phi$ *is a Markov representation if the following conditions hold for every timestep* $t$ *and any policy* $\pi$:

1. **Inverse Model.** *The inverse dynamic model, defined as* $I(a|s', s) := \frac{\mathcal{T}(s'|a,s)\pi(a|s)}{P^\pi(s'|s)}$, *where* $P^\pi(s'|s) = \sum_{\bar{a} \in \mathbb{A}} \mathcal{T}(s'|\bar{a}, s)\pi(a|s)$, *should be equal in the original and reduced MDPs. That is we have* $P^\pi(a|z', z) = P^\pi(a|s, s'), \forall a \in \mathbb{A}, s, s' \in \mathbb{S}$.

2. **Density Ratio.** *The original and abstract next-state density ratios are equal when conditioned on the same abstract state:* $\frac{P^\pi(z'|z)}{P^\pi(z')} = \frac{P^\pi(s'|z)}{P^\pi(s')}, \forall x' \in \mathbb{S}$, *where* $P^\pi(s'|z) = \sum_{\bar{s} \in \mathbb{S}} P^\pi(s'|\bar{s})\mu(\bar{s}|z)$ *and* $\mu(s|z) = \frac{\mathbf{1}_{\phi(s)=z} P^\pi(s)}{\sum_{\bar{s} \in \mathbb{S}} P^\pi(s|\bar{s})}$. $P^\pi(s'|z)$ *is the probability of transitioning to state* $s'$ *and* $\mu(s|z)$ *is the probability of currently being in state* $s$ *when in latent state* $z$.

We begin with two observations that are useful for our derivation.

Observation A: Given that any $z \in \mathbb{Z}$ is obtained from the mapping $x \xrightarrow{\Omega} o \xrightarrow{\phi} z$, and that $h \triangleq \phi \circ \Omega$ is a deterministic (but not necessarily invertible) function, each element $x \in \mathbb{X}$ maps to a single element $z \in \mathbb{Z}$. It directly follows that $\forall a, z_1, z_2 \in \mathbb{A} \times \mathbb{Z} \times \mathbb{Z}$, we have

$$p(a, z_1, z_2) = \sum_{x_1, x_2 \in \mathbb{X}^2} p(a, x_1, x_2)\mathbf{1}[z_1, z_2 = h(x_1), h(x_2)]$$

and

$$p(z_1, z_2) = \sum_{x_1, x_2 \in \mathbb{X}^2} p(x_1, x_2)\mathbf{1}[z_1, z_2 = h(x_1), h(x_2)]$$

Observation B: Let $P^\pi(a, x, x')$ be the joint distribution of elements in $(A, X, X')$ collected under policy $\pi$, we have $P^\pi(a, x_1, x_2) > 0$ if and only if $a, x_1, x_2 \in (A, X, X')$.

Observation C: Similarly to obs. B, we have $P^\pi(x_1, x_2) > 0$ if and only if $x_1, x_2 \in (X, X')$.

1) *Proving that the Inverse Model condition is satisfied if and only if* $I((Z, Z'); A) = I((X, X'); A)$.

The above is equivalent to showing that the Inverse Model condition is satisfied if and only if $\mathbf{H}(A|Z, Z') = \mathbf{H}(A|X, X')$. For $\mathbf{H}(A|Z, Z')$, we have

$$\mathbf{H}(A|Z, Z') = - \sum_{A,Z,Z'} P^\pi(a, z, z') \log P^\pi(a|z, z')$$

$$\text{(from obs. A)} \quad = - \sum_{\mathbb{A} \times \mathbb{Z} \times \mathbb{Z}} \sum_{x_1, x_2 \in \mathbb{X}^2} P^\pi(a, x_1, x_2) \mathbf{1}[z, z' = h(x_1), h(x_2)] \log P^\pi(a|z, z')$$

$$\text{(from obs. B)} \quad = - \sum_{\mathbb{A} \times \mathbb{X} \times \mathbb{X}} P^\pi(a, x, x') \sum_{\mathbb{Z}^2} \mathbf{1}[z, z' = h(x), h(x')] \log P^\pi(a|z, z')$$

$$= - \sum_{\mathbb{A} \times \mathbb{X} \times \mathbb{X}} P^\pi(a, x, x') \log \prod_{\mathbb{Z}^2} P^\pi(a|z, z')^{\mathbf{1}[z, z' = h(x), h(x')]}$$

$$= - \sum_{\mathbb{A} \times \mathbb{X} \times \mathbb{X}} P^\pi(x, x') P^\pi(a|x, x') \log \prod_{\mathbb{Z}^2} P^\pi(a|z, z')^{\mathbf{1}[z, z' = h(x), h(x')]}$$

$$= -\mathbb{E}_{X, X'}\Big[\sum_{\mathbb{A}} P^\pi(a|x, x') \log \prod_{\mathbb{Z}^2} P^\pi(a|z, z')^{\mathbf{1}[z, z' = h(x), h(x')]}\Big]$$

It follows that

$$\mathbf{H}(A|Z, Z') - \mathbf{H}(A|X, X') = \mathbb{E}_{X, X'}\Bigg[\sum_{\mathbb{A}} P^\pi(a|x, x') \log \frac{P^\pi(a|x, x')}{\prod_{\mathbb{Z}^2} P^\pi(a|z, z')^{\mathbf{1}[z, z' = h(x), h(x')]}}\Bigg]$$

$$= \mathbb{E}_{X, X'}[D_{\mathrm{KL}}(P\|Q)],$$

with $P = P^\pi(a|x, x')$ and $Q = \prod_{z,z' \in Z, Z'} P^\pi(a|z, z')^{\mathbf{1}[z, z' = h(x), h(x')]}$. From Gibbs inequality we always have $D_{\mathrm{KL}}(p\|q) \geq 0$, therefore $\mathrm{I}((Z, Z'); A) = \mathrm{I}((X, X'); A)$ if and only if $D_{\mathrm{KL}}(P\|Q) = 0$ $\forall x, x' \in X, X'$, which is the case if and only if $P = Q$ almost $\mu$-everywhere.

From observation A, any $x_1, x_2 \in \mathbb{X}^2$ maps to exactly one pair $z_1, z_2 \in \mathbb{Z}^2$, and by construction of $X, X', Z, Z'$, for any pair $x, x' \in X, X'$, we must have $Q = \prod_{\bar{z}, \bar{z}' \in \mathbb{Z}^2} P^\pi(a|\bar{z}, \bar{z}')^{\mathbf{1}[\bar{z}, \bar{z}' = h(x), h(x')]} = P^\pi(a|z, z')$, with $z, z'$ being the corresponding pair in $Z, Z'$.

Therefore $\mathrm{I}((Z, Z'); A) = \mathrm{I}((X, X'); A)$ if and only if $P^\pi(a|x, x') = P^\pi(a|z, z') \forall x, x', z, z' \in X, X', Z, Z'$, and we recover the Inverse Model condition.

Conversely, if the Inverse Model condition is not satisfied, then $\exists x, x', z, z', a \in X, X', Z, Z', A$ for which $P \neq Q$. Then $D_{\mathrm{KL}}(P\|Q) > 0$ at $x, x'$ and $\mathrm{I}((Z, Z'); A) < \mathrm{I}((X, X'); A)$.

2) *Proving that the Density Ratio condition is satisfied if* $\mathrm{I}(Z; Z') = \mathrm{I}(X; X')$.

We first show that satisfying

$$\frac{P^\pi(x'|x)}{P^\pi(x')} = \frac{P^\pi(z'|z)}{P^\pi(z')} \quad \forall x, x', z, z' \in X, X', Z, Z' \tag{11}$$

is sufficient for satisfying the Density Ratio condition $\frac{P^\pi(x'|z)}{P^\pi(x')} = \frac{P^\pi(z'|z)}{P^\pi(z')}$. We then show that the condition in Equation (11) holds if and only if $\mathrm{I}(Z; Z') = \mathrm{I}(X; X')$.

i) *Showing the Density Ratio condition holds when Equation* (11) *is satisfied.* First we notice that, $\forall x', z \in X', Z$, we have

$$P^\pi(x'|z) = \sum_{\bar{x} \in \mathbb{X}} \mathbf{1}[z = h(\bar{x})] P^\pi(x'|\bar{x}) = \mathbb{E}_X[P^\pi(x'|x)].$$

Then, supposing Equation (11) holds, we must have

$$P^\pi(x'|z) = \mathbb{E}_X[P^\pi(x'|x)] = P^\pi(x') \frac{P^\pi(z'|z)}{P^\pi(z')} \quad \forall x', z, z' \in X', Z, Z',$$

and the Density Ratio condition holds.

ii) *Proving Equation* (11) *holds if and only if* $\mathrm{I}(Z; Z') = \mathrm{I}(X; X')$.

We have

$$\mathrm{I}(Z; Z') = \sum_{\mathbb{Z}^2} P^\pi(z, z') \log \frac{P^\pi(z'|z)}{P^\pi(z')}$$

$$\text{(from obs. A)} \quad = \sum_{\mathbb{Z}^2} \sum_{x_1, x_2 \in \mathbb{X}^2} P^\pi(x_1, x_2) \mathbf{1}[z, z' = h(x_1), h(x_2)] \log \frac{P^\pi(z'|z)}{P^\pi(z')}$$

$$\text{(from obs. C)} \quad = \sum_{\mathbb{X}^2} P^\pi(x, x') \sum_{\mathbb{Z}^2} \mathbf{1}[z, z' = h(x), h(x')] \log \frac{P^\pi(z'|z)}{P^\pi(z')}$$

$$= \mathbb{E}_{X, X'} \left[ \log \prod_{\mathbb{Z}^2} \left( \frac{P^\pi(z'|z)}{P^\pi(z')} \right)^{\mathbf{1}[z, z' = h(x), h(x')]} \right].$$

Then,

$$\mathrm{I}(X; X') - \mathrm{I}(Z; Z') = \mathbb{E}_{X, X'}[D_{\mathrm{KL}}(P' \| Q')],$$

with

$$P' = \frac{P^\pi(x'|x)}{P^\pi(x')} \quad \text{and} \quad Q = \prod_{\mathbb{Z}^2} \left( \frac{P^\pi(z'|z)}{P^\pi(z')} \right)^{\mathbf{1}[z, z' = h(x), h(x')]}$$

The remainder of this part follows the same structure as for the first part of the proof.

$\mathrm{I}(X; X') = \mathrm{I}(Z; Z')$ if and only if $\forall x, x' \in X, X', P = Q$ almost $\mu$-everywhere. Any $x_1, x_2 \in \mathbb{X}^2$ maps to exactly one pair $z_1, z_2 \in \mathbb{Z}^2$, and by construction of $X, X', Z, Z'$, for any pair $x, x' \in X, X'$, we must have

$$Q = \prod_{\bar{z}, \bar{z}' \in \mathbb{Z}^2} \left( \frac{P^\pi(\bar{z}'|\bar{z})}{P^\pi(\bar{z}')} \right)^{\mathbf{1}[\bar{z}, \bar{z}' = h(x), h(x')]} = \frac{P^\pi(z'|z)}{P^\pi(z')},$$

with $z, z'$ being the corresponding pair in $Z, Z'$.

Therefore $\mathrm{I}(X; X') = \mathrm{I}(Z; Z')$ if and only if $\forall x, x', z, z' \in X, X', Z, Z'$ we have $\frac{P^\pi(x'|x)}{P^\pi(x')} = \frac{P^\pi(z'|z)}{P^\pi(z')}$. Finally, from i) being true, the Density ratio condition must hold. $\square$

**Lemma 4.1.** $I(Z; L) > 0$ if $I(O; L) > 0$ and $\exists z_k, c_j \in Z \times L$ such that $\mu(z_k|c_j) \neq \mu(z_k)$.

*Proof.* Given $\pi$ is fixed while the batch $O$ is collected, for a single batch the causal interaction between $L, O$ and $Z$ is described by the Markov chain $X \to O \to Z$, where $x = (s, c) \in \mathbb{S} \times L$ and isn't directly observed. By the data processing inequality, $\mathrm{I}(L; Z) \leq \mathrm{I}(L; O)$, and as such $\mathrm{I}(L; O) > 0$ is a necessary condition for $\mathrm{I}(L; Z)$ to be positive.

Note that

$$\mathrm{I}(L; Z) = \mathbf{H}(L) + \mathbf{H}(Z) - \mathbf{H}(L, Z) = 0 \Leftrightarrow \mathbf{H}(L, Z) = \mathbf{H}(L) + \mathbf{H}(Z),$$

that is, if and only if $Z$ and $L$ are independently distributed. Given the causal relationship between $L$ and $Z$, $\mu(z|c)$ is well defined $\forall z, c \in Z \times L$. If $\exists z_k, c_j \in Z \times L$ such that $\mu(z_k|c_j) \neq \mu(z_k)$ then $Z$ and $L$ cannot be independently distributed, and $\mathrm{I}(L; Z) > 0$. $\square$

**Lemma 4.2.** $I(Z; L)$ *monotonically increases with a)* $I(Z; V) - I(Z; V|L)$ *and b)* $I(Z; Z') - I(Z; Z'|L)$.

*Proof.* Proof for condition a) : By the causal structure $V \leftarrow X \to O \to Z$ and the chain rule of mutual information, we have

$$\mathrm{I}(Z; L) = \mathrm{I}(Z; V) - \mathrm{I}(Z; V|L) + \mathrm{I}(Z; L|V),$$

$\mathrm{I}(Z; L|V)$ is the information encoded in $Z$ about the training levels that does not depend on $V$. $\mathrm{I}(Z; V) - \mathrm{I}(Z; V|L)$ represents the information encoded in $Z$ about state values that is level-specific. If this term increases then $\mathrm{I}(Z; L)$ will also increase.

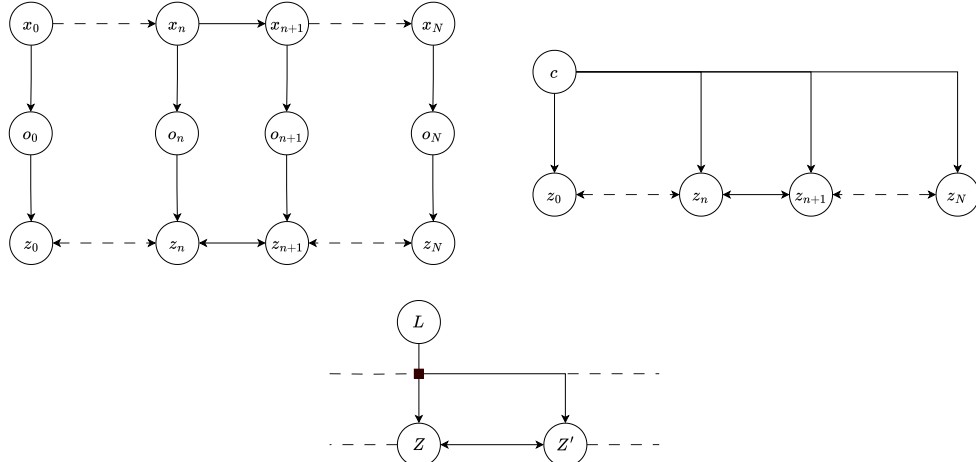

Figure 6: In the top row, left, we depict the causal graph of states, observation and latents obtained over an episode. On the same row we draw a simplified graph that focuses on the relationship between $c$ and $Z_{0:N}$, and utilises the notion that the context remains the same throughout the episode. In the bottom row we draw the resulting causal relationship between $L$, $Z$ and $Z'$.

Proof for condition b) : Consider an episode of arbitrary length $N$ collected with policy $\pi$. We depict the causal structure that exists between elements in the top row of Figure 6 (elements may be repeated within each sequence). It naturally follows that we have the causal structure depicted in the bottom row when considering all levels in $L$. By the chain rule of mutual information, we have

$$\mathrm{I}(Z;L) = \mathrm{I}(Z;(Z',L)) - \mathrm{I}(Z;Z'|L) = \mathrm{I}(Z;L|Z') + \mathrm{I}(Z;Z') - \mathrm{I}(Z;Z'|L),$$

and it follows that $\mathrm{I}(Z;L)$ increases with $\mathrm{I}(Z;Z') - \mathrm{I}(Z;Z'|L)$.

$\square$

# B  ADDITIONAL FIGURES AND TABLES

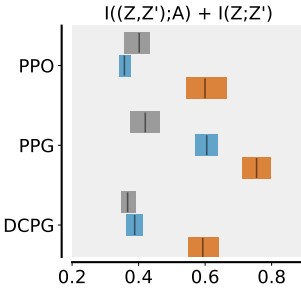

Figure 7: $I((Z,Z');A) + I(Z;Z')$ for shared (gray), actor (blue) and critic (orange) for PPO, PPG, and DCPG in Procgen.

# C  IMPLEMENTATION DETAILS

## C.1  MUTUAL INFORMATION ESTIMATION

We measure mutual information using the estimator proposed by Kraskov et al. (2004) and later extended to pairings of continuous and discrete variables by Ross (2014). These methods are based

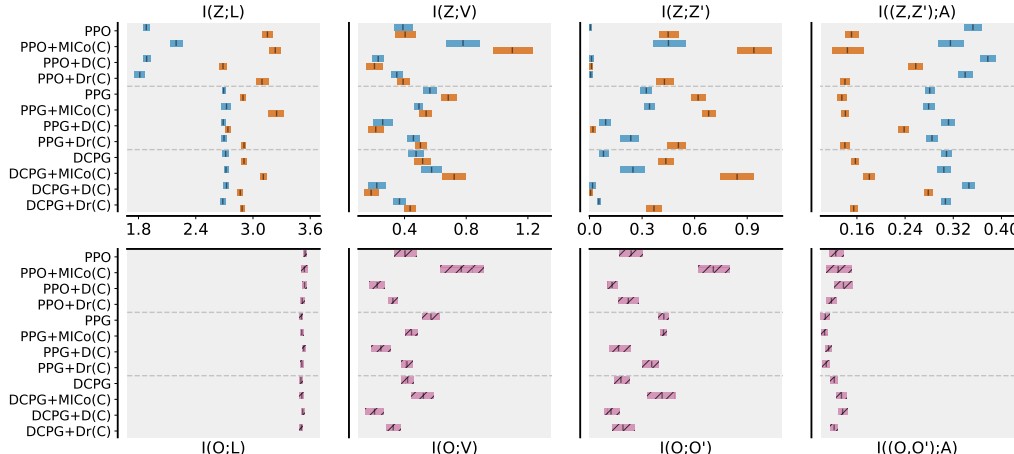

Figure 8: Mutual information measurements for the actor (blue) and critic (orange) for auxiliary losses applied to the critic for PPO, PPG, and DCPG in Procgen. Top/bottom rows are $I(Z; \cdot)/I(O; \cdot)$ with a shared x-axis.

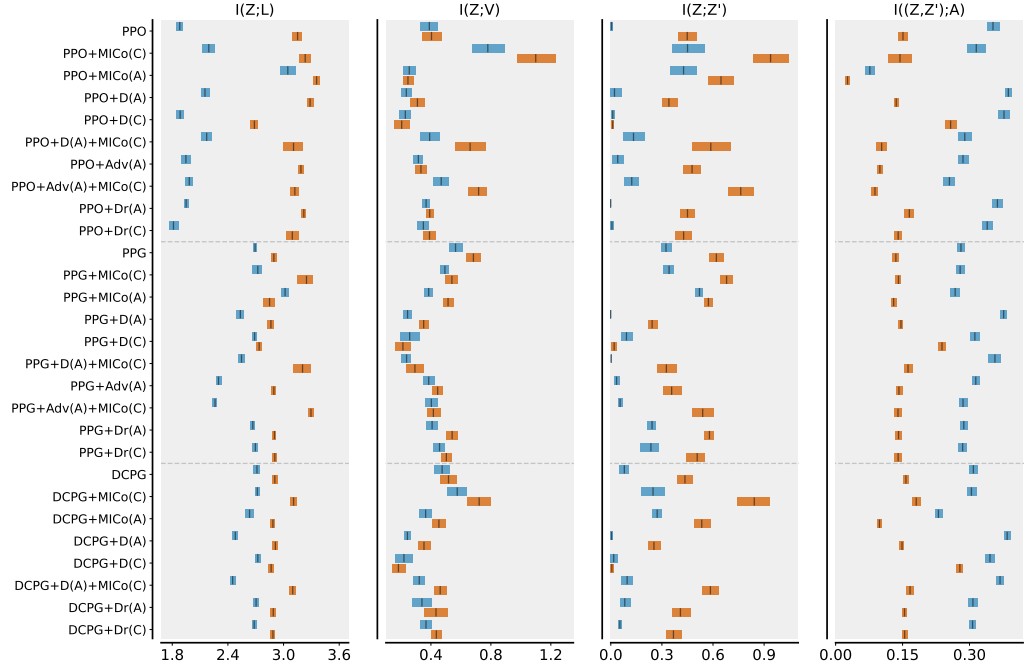

Figure 9: $I(Z; \cdot)$ measurements for the actor (blue) and critic (orange) for auxiliary losses for PPO, PPG, and DCPG in Procgen.

on performing entropy estimation using k-nearest neighbors distances. We use $k = 3$ and determine nearest neighbors by measuring the Euclidian ($L_2$) distance between points. We checked measurements obtained when using different $k$ and under different metric spaces, and we found that our measurements are broadly invariant to the choice of estimator parameters.

At the end of training we collect a batch of trajectories consisting of $2^{16}$ timesteps ($2^{15}$ timesteps in Brax) from $L$. We construct $(A, O, O', Z, Z', V, L)$ from $n = 4096$ timesteps yielding $(a_t, o_t, o_{t+1}, z_t, z_{t+1}, v_t, c_t)$. Subsampling is necessary to compute mutual information estimates in a reasonable time, while ensuring we sample states from most levels in $L$ and at various point of

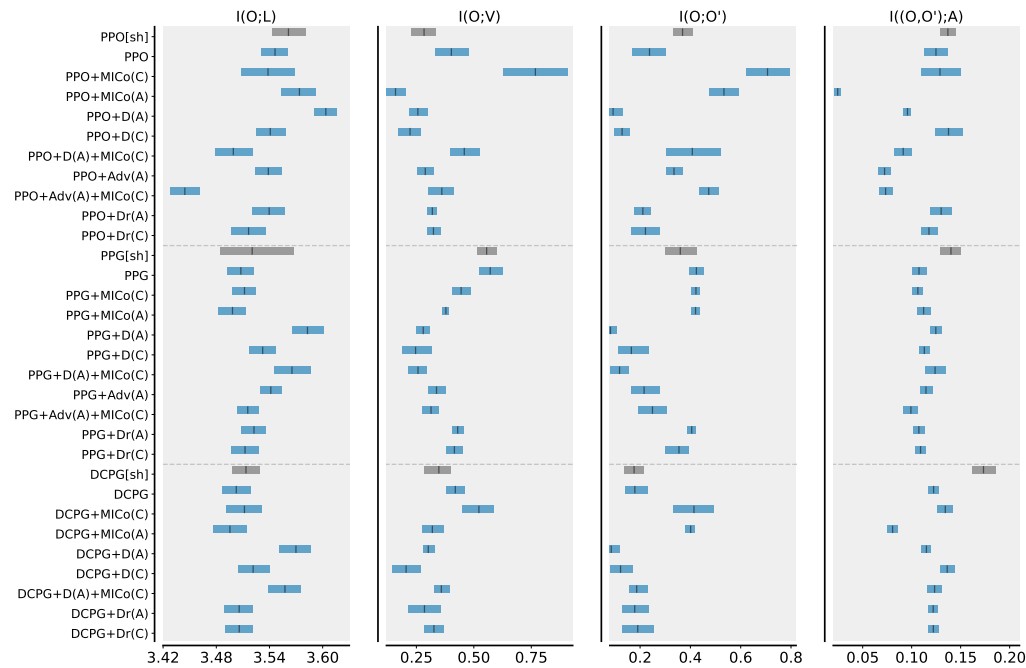

Figure 10: $I(O; \cdot)$ measurements for the actor (blue) and critic (orange) for auxiliary losses for PPO, PPG, and DCPG in Procgen.

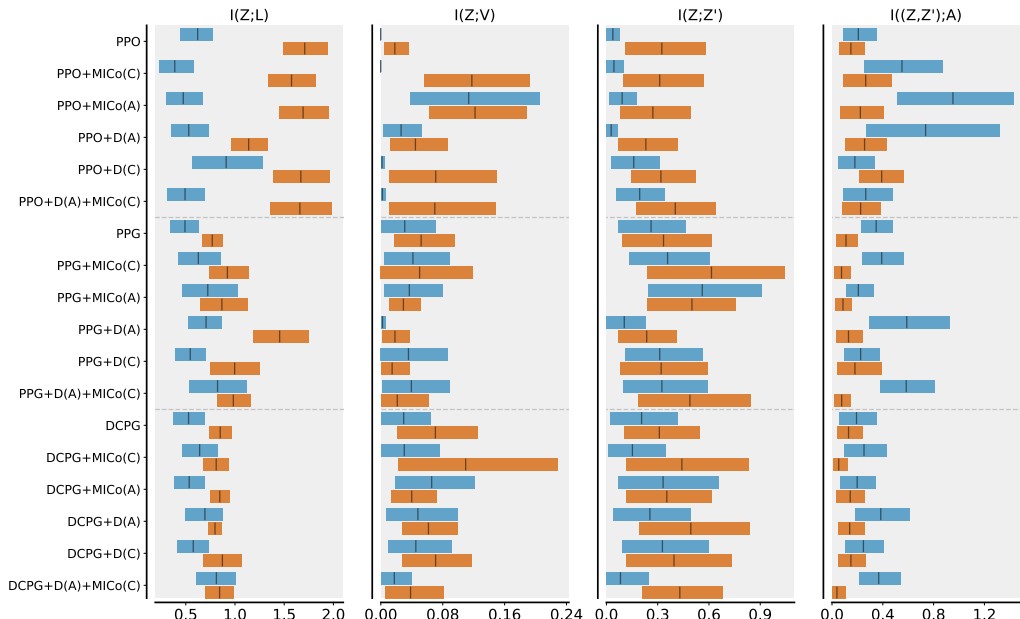

Figure 11: Mutual information measurements for the actor (blue) and critic (orange) for auxiliary losses for PPO, PPG, and DCPG in Brax.

the trajectories followed by the agent in each level. Timesteps are sampled uniformly and without replacement from the batch, after having excluded:

1. Odd timesteps, to ensure $O$ and $O'$ will not overlap (i.e. $O$ contains only even timesteps, and $O'$, being sampled at $t + 1$, contains only odd timesteps).

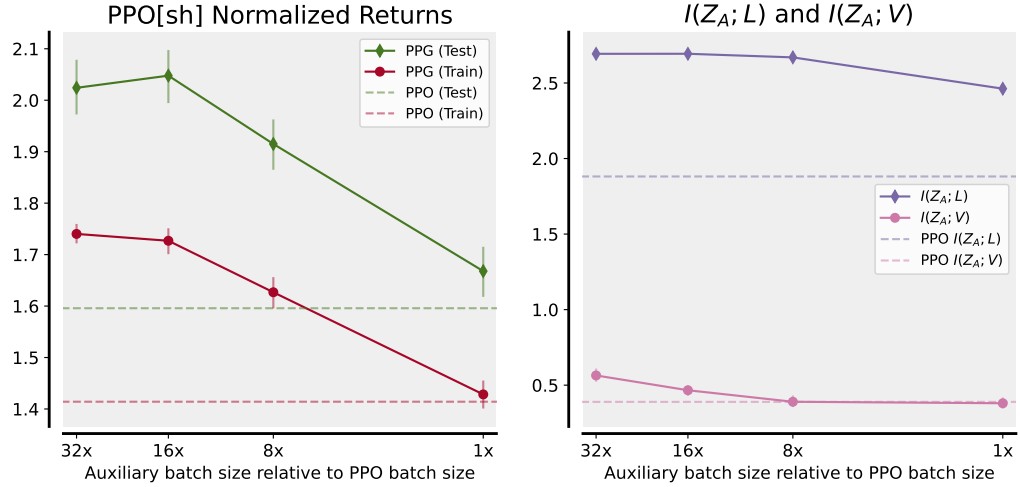

Figure 12: Procgen PPG returns (left) normalized by PPO[sh] performance and mutual information quantities $I(Z_A; L)/I(Z_A; V)$ (right) for varying auxiliary batch size levels.

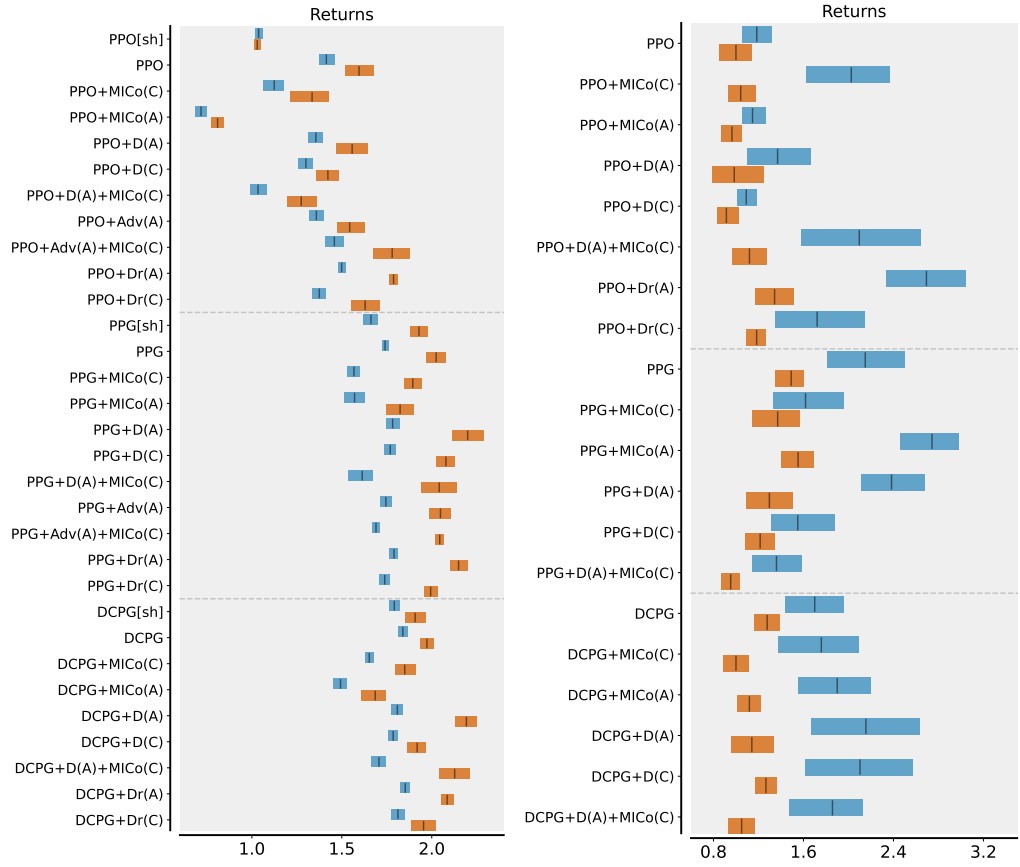

Figure 13: Returns in Procgen (left) and Brax (right).

2. Timesteps corresponding to episode terminations, to ensure the pair $o_t, o_t + 1$ cannot originate from different levels.

3. Timesteps from episodes that have not terminated, to ensure we can always compute $v_t$.

Table 2: Measurements of compression efficiency C($Z_A|O;V$) (Equation (9)) with standard error in Procgen. Statistical significance bolded, determined by Welch's t-test. Results highlighted in red when decoupling decreases C($Z_A|O;V$), and highlighted in green when decoupling increases C($Z_A|O;V$), otherwise yellow. Coupled architectures are denoted with algorithm name plus "[sh]".

| Algorithm | $C(Z_A|O;V)$ | $C(Z_A|O;L)$ |
|-----------|--------------|--------------|
| PPO[sh] | $89.3 \pm 2$ | $65.2 \pm 3$ |
| PPO | $90.1 \pm 4$ | $\mathbf{52.3 \pm 3}$ |
| PPG[sh] | $85.9 \pm 4$ | $70.0 \pm 2$ |
| PPG | $94.1 \pm 2$ | $\mathbf{75.5 \pm 2}$ |
| DCPG[sh] | $95.4 \pm 2$ | $77.6 \pm 2$ |
| DCPG | $92.3 \pm 7$ | $76.4 \pm 2$ |

Table 3: Measurements of compression efficiency C($Z_A|O;\cdot$) (Equation (9)) of the actor's representation $\phi_A$ in Procgen. Results highlighted in red when the auxiliary loss decreases the metric relative to the base algorithm, and highlighted in green when the auxiliary loss increases the metric relative to the base algorithm. Auxiliary losses are applied to the actor (A) and critic (C) in the form of dynamics prediction (D), MICo, and advantage distillation (Adv).

| Algorithm | $C(Z_A|O;V)$ | $C(Z_A|O;L)$ | $C((Z_A|O, Z'_A|O'); A)$ |
|-----------|--------------|--------------|---------------------------|
| PPO | $90.1 \pm 4$ | $52.3 \pm 3$ | $99.9 \pm 0$ |
| PPO+MICo(C) | $93.9 \pm 2$ | $\mathbf{60.4 \pm 3}$ | $99.4 \pm 0$ |
| PPO+MICo(A) | $98.6 \pm 1$ | $\mathbf{84.7 \pm 2}$ | $\mathbf{87.5 \pm 5}$ |
| PPO+D(A) | $62.6 \pm 12$ | $\mathbf{61.3 \pm 2}$ | $100.0 \pm 0$ |
| PPO+D(C) | $86.8 \pm 7$ | $52.8 \pm 3$ | $100.0 \pm 0$ |
| PPO+D(A)+MICo(C) | $76.3 \pm 6$ | $\mathbf{63.7 \pm 3}$ | $99.5 \pm 0$ |
| PPO+Adv(A) | $96.9 \pm 2$ | $53.2 \pm 3$ | $100.0 \pm 0$ |
| PPO+Adv(A)+MICo(C) | $\mathbf{100.0 \pm 0}$ | $57.0 \pm 2$ | $98.5 \pm 1$ |
| PPO+Dr(A) | $89.1 \pm 6$ | $54.3 \pm 3$ | $100.0 \pm 0$ |
| PPO+Dr(C) | $98.1 \pm 1$ | $50.8 \pm 3$ | $99.6 \pm 0$ |
| PPG | $94.1 \pm 2$ | $75.5 \pm 2$ | $100.0 \pm 0$ |
| PPG+MICo(C) | $98.0 \pm 1$ | $76.4 \pm 2$ | $100.0 \pm 0$ |
| PPG+MICo(A) | $95.4 \pm 2$ | $\mathbf{85.8 \pm 2}$ | $100.0 \pm 0$ |
| PPG+D(A) | $89.5 \pm 4$ | $71.2 \pm 2$ | $100.0 \pm 0$ |
| PPG+D(C) | $96.3 \pm 2$ | $75.1 \pm 2$ | $100.0 \pm 0$ |
| PPG+D(A)+MICo(C) | $85.9 \pm 7$ | $71.6 \pm 2$ | $100.0 \pm 0$ |
| PPG+Adv(A) | $98.4 \pm 1$ | $\mathbf{63.3 \pm 2}$ | $100.0 \pm 0$ |
| PPG+Adv(A)+MICo(C) | $99.4 \pm 1$ | $\mathbf{62.9 \pm 2}$ | $100.0 \pm 0$ |
| PPG+Dr(A) | $91.3 \pm 3$ | $74.9 \pm 2$ | $100.0 \pm 0$ |
| PPG+Dr(C) | $93.1 \pm 6$ | $75.6 \pm 2$ | $100.0 \pm 0$ |
| DCPG | $92.3 \pm 7$ | $76.4 \pm 2$ | $100.0 \pm 0$ |
| DCPG+MICo(C) | $98.1 \pm 1$ | $76.6 \pm 2$ | $100.0 \pm 0$ |
| DCPG+MICo(A) | $91.7 \pm 3$ | $74.3 \pm 2$ | $100.0 \pm 0$ |
| DCPG+D(A) | $80.9 \pm 4$ | $\mathbf{69.9 \pm 2}$ | $100.0 \pm 0$ |
| DCPG+D(C) | $97.8 \pm 1$ | $76.3 \pm 2$ | $100.0 \pm 0$ |
| DCPG+D(A)+MICo(C) | $83.4 \pm 5$ | $\mathbf{69.6 \pm 2}$ | $100.0 \pm 0$ |
| DCPG+Dr(A) | $97.5 \pm 2$ | $76.7 \pm 2$ | $100.0 \pm 0$ |

## C.2 PROCGEN

The Procgen Benchmark is a set of 16 diverse PCG environments that echoes the gameplay variety seen in the ALE benchmark Bellemare et al. (2015). The game levels, determined by a random seed,

Table 4: Measurements of compression efficiency $C(Z_C|O; \cdot)$ (Equation (9)) of the actor's representation $\phi_C$ in Procgen. Results highlighted in red when the auxiliary loss decreases the metric relative to the base algorithm, highlighted in green when the auxiliary loss increases the metric relative to the base algorithm, and highlighted in yellow otherwise. Auxiliary losses are applied to the actor (A) and critic (C) in the form of dynamics prediction (D), MICo, and advantage distillation (Adv).

| Algorithm | $C(Z_C|O; V)$ | $C(Z_C|O; L)$ | $C((Z_C|O, Z'_C|O'; A))$ |
|---|---|---|---|
| PPO | $93.7 \pm 3$ | $88.4 \pm 2$ | $85.6 \pm 4$ |
| PPO+MICo(C) | $100.0 \pm 0$ | $90.3 \pm 1$ | $82.7 \pm 3$ |
| PPO+MICo(A) | $97.6 \pm 2$ | $92.4 \pm 1$ | $\mathbf{64.4 \pm 6}$ |
| PPO+D(A) | $99.7 \pm 0$ | $90.2 \pm 1$ | $87.4 \pm 3$ |
| PPO+D(C) | $87.6 \pm 4$ | $\mathbf{77.4 \pm 2}$ | $\mathbf{99.1 \pm 0}$ |
| PPO+D(A)+MICo(C) | $91.0 \pm 6$ | $88.1 \pm 2$ | $84.4 \pm 3$ |
| PPO+Adv(A) | $96.7 \pm 2$ | $89.3 \pm 1$ | $87.6 \pm 4$ |
| PPO+Adv(A)+MICo(C) | $100.0 \pm 0$ | $89.9 \pm 1$ | $87.0 \pm 3$ |
| PPO+Dr(A) | $98.0 \pm 1$ | $90.0 \pm 1$ | $87.9 \pm 3$ |
| PPO+Dr(C) | $97.5 \pm 1$ | $87.0 \pm 2$ | $86.3 \pm 3$ |
| PPG | $99.2 \pm 1$ | $81.6 \pm 2$ | $90.3 \pm 3$ |
| PPG+MICo(C) | $93.0 \pm 6$ | $\mathbf{90.9 \pm 2}$ | $91.8 \pm 2$ |
| PPG+MICo(A) | $100.0 \pm 0$ | $80.9 \pm 2$ | $84.4 \pm 4$ |
| PPG+D(A) | $100.0 \pm 0$ | $79.2 \pm 2$ | $91.3 \pm 3$ |
| PPG+D(C) | $\mathbf{89.4 \pm 4}$ | $77.6 \pm 2$ | $\mathbf{100.0 \pm 0}$ |
| PPG+D(A)+MICo(C) | $93.3 \pm 4$ | $\mathbf{89.1 \pm 2}$ | $87.1 \pm 4$ |
| PPG+Adv(A) | $100.0 \pm 0$ | $80.9 \pm 2$ | $89.7 \pm 3$ |
| PPG+Adv(A)+MICo(C) | $99.9 \pm 0$ | $92.3 \pm 1$ | $93.1 \pm 3$ |
| PPG+Dr(A) | $98.7 \pm 1$ | $81.7 \pm 2$ | $90.2 \pm 3$ |
| PPG+Dr(C) | $96.4 \pm 3$ | $81.8 \pm 2$ | $85.8 \pm 4$ |
| DCPG | $99.5 \pm 1$ | $81.7 \pm 2$ | $92.1 \pm 3$ |
| DCPG+MICo(C) | $98.9 \pm 1$ | $\mathbf{88.6 \pm 2}$ | $93.5 \pm 2$ |
| DCPG+MICo(A) | $99.8 \pm 0$ | $81.4 \pm 2$ | $87.2 \pm 4$ |
| DCPG+D(A) | $100.0 \pm 0$ | $80.7 \pm 2$ | $92.6 \pm 3$ |
| DCPG+D(C) | $\mathbf{91.3 \pm 3}$ | $81.2 \pm 1$ | $\mathbf{100.0 \pm 0}$ |
| DCPG+D(A)+MICo(C) | $99.6 \pm 0$ | $\mathbf{87.4 \pm 2}$ | $92.7 \pm 3$ |
| DCPG+Dr(A) | $100.0 \pm 0$ | $81.3 \pm 2$ | $89.4 \pm 3$ |
| DCPG+Dr(C) | $97.0 \pm 2$ | $81.2 \pm 2$ | $90.7 \pm 3$ |

can differ in visual design, navigational structure, and the starting locations of entities. All Procgen environments use a common discrete 15-dimensional action space and generate $64 \times 64 \times 3$ RGB observations. A detailed description of each of the 16 environments is provided by Cobbe et al. (2020). RL algorithms such as PPO reveal significant differences between test and training performance in all games, making Procgen a valuable tool for evaluating generalisation performance.

We conduct our experiment on the easy setting of Procgen, which employs 200 training levels and a budget of 25M training steps, and evaluate the agent's scores on the training levels and on the full range of levels, excluding the training levels. We use the version of Procgen provided by EnvPool (Weng et al., 2022). Following prior work, (Raileanu et al., 2021; Jiang et al., 2021; Moon et al., 2022), for each game we normalise train/test scores by the mean train/test score achieved by PPO in that game.

For PPO, we base our implementation on the CleanRL PPO implementation (Huang et al., 2022), which reimplements the PPO agent from the original Procgen publication in JAX. We use the same

ResNet policy architecture and PPO hyperparameters (identical for all games) as Cobbe et al. (2020) and reported in Table 5.

We re-implement PPG and DCPG in JAX, based on the Pytorch implementations provided by Huang et al. (2022) and Moon et al. (2022). We use the default recommended hyperparameters for each algorithms, which are reported in Table 6. We note that our PPG implementation ends up outperforming the original implementation by about 10% on the test set, while our DCPG implementation underperforms test scores reported by Moon et al. (2022) by about 10%. We attribute this discrepancy to minor differences between the JAX and Pytorch libraries, and decided to not investigate further.

We conduct our experiments on A100 and RTX8000 Nvidia GPUs and 6 CPU cores. One seed for one game completes in 2 to 12 hours, depending on the GPU, algorithm, and whether the architecture is coupled or decoupled (for example, PPG decoupled can be expected to run 4x to 6x slower than PPO coupled).

### C.3 BRAX

For our experiments in Brax, we implement a custom "video distractors" set of tasks, similar to those from (Stone et al., 2021). In this setup, a video plays in an overlay on the pixels the agent views. There is a disjoint set of videos between the training and testing environments. The random seed determines the environment's initial physics and the video overlay at the beginning of training. The pixels themselves are full-RGB $64 \times 64 \times 3$ arrays, but we use framestacking to bring each agent input to $64 \times 64 \times 9$ pixels.

Similar to the algorithms used in the Procgen experiments, we implement our algorithms in JAX and base them on ClearnRL.

We conduct our experiments on RTX A4500 Nvidia GPUs and 6 CPU cores. One seed completes in 7.5-48 hours, depending on the environment and its physics backend as well as the algorithm.

Table 5: Hyperparameters used for PPO in Procgen and Brax experiments. All runs employing a specific (or combination of) representation learning objective use the same hyperparameters.

| Parameter | Procgen | Brax |
|---|---|---|
| *PPO* | | |
| $\gamma$ | 0.999 | 0.999 |
| $\lambda_{\text{GAE}}$ | 0.95 | 0.95 |
| rollout length | 256 | 128 |
| minibatches per epoch | 8 | 8 |
| minibatch size | 2048 | 512 |
| $J_\pi$ clip range | 0.2 | 0.2 |
| number of environments | 64 | 32 |
| Adam learning rate | 5e-4 | 5e-4 |
| Adam $\epsilon$ | 1e-5 | 1e-8 |
| max gradient norm | 0.5 | 0.5 |
| value clipping | no | no |
| return normalisation | yes | no |
| value loss coefficient | 0.5 | 0.5 |
| entropy coefficient | 0.01 | 0.01 |
| | | |
| *PPO (coupled)* | | |
| PPO epochs (actor and critic) | 3 | - |
| | | |
| *PPO (decoupled)* | | |
| Actor epochs | 1 | 1 |
| Critic epochs | 9 | 1 |
| | | |
| *MICo objective* | | |
| MICo coefficient | 0.5 | 0.01 |
| Target network update coefficient | 0.005 | 0.05 |
| | | |
| *Dynamics objective* | | |
| Dynamics loss coefficient | 1.0 | 0.01 |
| In-distribution transitions weighting | 1.0 | 1.0 |
| Out-of-distribution states weighting | 1.0 | 1.0 |
| Out-of-distribution actions weighting | 0.5 | 0.5 |
| | | |
| *Advantage distillation objective* | | |
| Advantage prediction coefficient | 0.25 | - |

Table 6: Hyperparameters used for PPG and DCPG in Procgen experiments. Hyperparameters shared between methods are only reported if they change from the method above. All runs employing a specific (or combination of) representation learning objective use the same hyperparameters.

| Parameter | Procgen |
|---|---|
| *PPG* | |
| $\gamma$ | 0.999 |
| $\lambda_{\text{GAE}}$ | 0.95 |
| rollout length | 256 |
| minibatches per epoch policy phase | 8 |
| minibatch size policy phase | 2048 |
| minibatches per epoch auxiliary phase | 512 |
| minibatch size auxiliary phase | 1024 |
| $J_\pi$ clip range | 0.2 |
| number of environments | 64 |
| Adam learning rate | 5e-4 |
| Adam $\epsilon$ | 1e-5 |
| max gradient norm | 0.5 |
| value clipping | no |
| return normalisation | yes |
| value loss coefficient policy phase | 0.5 |
| value loss coefficient auxiliary phase | 1.0 |
| entropy coefficient | 0.01 |
| policy phase epochs | 1 |
| auxiliary phase epochs | 6 |
| number of policy phases per auxiliary phase | 32 |
| policy regularisation coefficient $\beta_c$ | 1.0 |
| auxiliary value distillation coefficient | 1.0 |
| | |
| *DCPG* | |
| value loss coefficient policy phase | 0.0 |
| delayed value loss coefficient policy phase | 1.0 |
| | |
| *MICo objective* | |
| MICo coefficient | 0.5 |
| Target network update coefficient | 0.005 |
| | |
| *Dynamics objective* | |
| Dynamics loss coefficient | 1.0 |
| In-distribution transitions weighting | 1.0 |
| Out-of-distribution states weighting | 1.0 |
| Out-of-distribution actions weighting | 0.5 |

