# OpenReview forum: "Studying the Interplay Between the Actor and Critic Representations in Reinforcement Learning"
_ICLR.cc/2025/Conference — ICLR 2025 Poster_

### Official Review · Reviewer_r4WB · 2024-10-27

**Soundness:** 3
**Presentation:** 2
**Contribution:** 3
**Rating:** 8
**Confidence:** 3

**Summary:**

Authors systematically investigate the impact of coupling and decoupling of the representations for actors and critics in on-policy actor-critic framework.

**Strengths:**

1. Theoretically-grounded study with the demonstration that actor and critic should be decoupled as their objectives contradict each other.
2. Good empirical study for confirming claims using various algorithms. When result contradict assumptions, authors test and confirm different hypothesis which could explain the observations. These assumptions reveal novel things for RL community, such as the potentially underestimated role of critic in exploration.
3. Additional study on the effects of different representation learning approaches and when they benefit performance.

**Weaknesses:**

1. Study is only conducted using the on-policy algorithms. It would be interesting to also investigate off-policy methods
2. If I understand correctly, study is performed only on discrete state and action spaces environments. Would the conclusions change if we switch to the continuous environments? It is not clear for me whether the proposed analysis can be applied to such tasks as the computation of mutual information becomes intractable.
3. For me, it was easy to get lost in many different metrics and keeping in mind what we want to maximize and what we want to minimize for the optimal solutions.

**Questions:**

My questions are based on the weaknesses

1. Could authors please elaborate on weaknesses 1 and 2?
2. According to weakness 3, I would recommend authors to add some additional information to the tables and plots about whether the metric is expected to be minimized or maximized.

---

> ### Author Response · Authors · 2024-11-23
> **Authors' response to review**
>
> We thank the reviewer for their insightful review. In regards to weaknesses 1 and 2, we kindly point the reviewer to the comment titled [proposed changes to manuscript] (at the top).
>
> In regards to weakness 3, we agree with the reviewer that keeping track of the different metrics can be difficult. In light of this, we are reworking Figure 1 (right) as a standalone table to contain compact definitions for each of the metrics, alongside the existing information reporting the changes observed in $\phi_A$ and $\phi_C$ once the representations are decoupled. A preview for this table can be accessed at https://ibb.co/vwHnfxS
>
> We hope the new table can act as a "cheatsheet" for the reader throughout the paper, reminding them how to intuitively understand each metric, and what they say about the representation.

---

> > ### Comment · Reviewer_r4WB · 2024-11-25
> >
> > I thank authors for their response and I increase my score

---

### Official Review · Reviewer_5hRY · 2024-11-04

**Soundness:** 4
**Presentation:** 4
**Contribution:** 3
**Rating:** 8
**Confidence:** 3

**Summary:**

This paper studies whether actors and critics benefit from a decoupled representation. The authors find that when decoupled, actors and critics learn representations that specialize in different types of information from the environment. Additionally, there is an interplay between these representations even when they are decoupled. The authors show how this insight helps select representation learning objectives for better performance.

**Strengths:**

**originality**
- although the fact that decoupled actor and critic learn different representations is not really surprising, the other findings in the paper are very interesting and novel.
-  the four mutual information metrics are interesting and helps to understand the effect of decoupled learning.
- the paper gives some very novel and interesting analysis and results

**quality**
- it is good that authors plan to release all data and code for better reproducibility.
- overall great quality

**clarity**
- writing is clear and easy to follow.

**significance**
- empirical and theoretical results are good contributions
- the analysis and insights from the paper are especially interesting

**Weaknesses:**

I don't have major concerns, minor ones:
- I think it can be nice to explicitly highlight, either in the beginning or conclusion, what are the best representation learning objectives and most important specialisations with quantitative measures.
- The paper can be even stronger if similar analysis can be done for Q-learning based methods such as DDPG/SAC/TD3 type algorithms. But I understand that might be too much extra work.

**Questions:**

- For the mutual information metrics, e.g., Figure 4, how much will they change if we simply change some of the hyperparameters of one algorithm? I am curious about factors other than coupled/decoupled learning and how much impact they can have on these metrics.

---

> ### Author Response · Authors · 2024-11-23
> **Authors' response to review**
>
> We thank the reviewer for their insightful review. We were glad to learn that the reviewer found the contributions made in this article interesting. We believe our comment titled [proposed changes to manuscript] (at the top) covers the weaknesses pointed out by the reviewer. We will therefore use this comment to answer the reviewer's question.
>
> > For the mutual information metrics, e.g., Figure 4, how much will they change if we simply change some of the hyperparameters of one algorithm?
>
> While we lack the computational budget to conduct a sensitivity analysis on all the possible hyperparameters, the additional experiments we are currently running should shed some light on the effect of changing the model capacity (i.e. the number of learned parameters) and the learning rate.
>
> We occasionally had to tune the weighting coefficient of the representation learning objectives tested to improve learning stability. The impact of this weighting coefficient on the mutual information metrics was as expected: If objective $X$ with coefficient $c_X$ increases information metric $I_A$ and decreases $I_B$, then increasing $c_X$ further increases $I_A$ and decreases $I_B$.

---

> > ### Comment · Reviewer_5hRY · 2024-11-27
> > **Thank you for your response**
> >
> > Thank you for answering my questions, I do not have other major concerns at this point.

---

### Official Review · Reviewer_qthx · 2024-11-06

**Soundness:** 2
**Presentation:** 1
**Contribution:** 2
**Rating:** 6
**Confidence:** 3

**Summary:**

This paper analyzes how different choices of architecture and representation learning objectives affect overfitting on, transfer to, and specializing in different levels of games (ProcGen). The analysis is largely established on the mutual information between different quantities. The author concludes that actor and critic representation specialize in different information, that critic can influence exploration significantly, and that representation learning objectives (i.e., auxiliary loss) should be consistent with the specialization of actor and critic representation.

**Strengths:**

The analysis to representation learning based on mutual information is interesting. The conclusion drawn from this analysis is helpful for deep RL researchers on designing representation learning objectives.

**Weaknesses:**

Writing is very unclear. See my questions for details. Overall, there are overwhelming notations, making the presentation unnecessarily complex. The reason of defining each notation is also not clear to me. Takeaway messages in each section is also not clear. For example, in section 4.1, the author starts with lots of new definitions without explaining why these are necessary. I would suggest the author explain why we need those definitions and then say the high-level messages before diving into details.

**Questions:**

- Line 154: Why measuring how easy it is to infer the identity of the current level from Z is a good measure of overfitting?
- Line 158: The purpose of Theorem 3.1 is not explained. Also, is it a result from the other paper or proved first time in this paper?
- Line 165: Why does a high I(Z_C, V) help optimizing the value loss?
- Line 166: What do you mean by increasing I(Z_A, V) and what's the relationship between value distillation and increasing I(Z_A, V)?
- Line 175: What do you mean by "whether it is possible to identify latents (ϕ(o), ϕ(o′))) obtained from consecutive observations from pairs stemming from non-consecutive observations?" Is it consecutive or not?
- Line 194: What's the role of linear probing here?
- Equation 6: What are $z^*_{A_0}$ and  $z^*_{A_1}$? Why I((Z^*_A, Z^*'_A); A) is guaranteed to be maximized when following the optimal policy?
- Line 256: I don't see why zero-shot transfer is evident in your example.
- Figure 3: How should one interpret these values? The higher the better? The lower the better? Better summarize the takeaway in the caption.
- Line 374: Why does maximizing J^\pi bias \pi to collect trajectories with high I(O; V)?
- Line 431: Why does actor objective promote learning invariant quantity?

---

> ### Author Response · Authors · 2024-11-23
> **Authors' response to review (1/2)**
>
> We thank the reviewer for their time and comments, and appreciate the reviewer’s suggestions. We believe incorporating these suggestions and addressing these question will improve the clarity of the presentation. We will take these suggestions into account in the new manuscript revision. We point the reviewer to the comment titled [proposed changes to manuscript] (at the top) for a summary of the changes we plan to make for the next revision. We hope these clarifications will make the reviewer consider improving their score.
>
> We will use this thread to address the reviewer’s questions.
>
> > Line 154: Why measuring how easy it is to infer the identity of the current level from Z is a good measure of overfitting?
>
> Being able to infer the identity of the training level from $Z$ implies the learned latent features are level-specific, and that $I(Z;L)$ is high.
>
> To help build an intuition on the relationship between $I(Z_A;L)$ and overfitting, we link to this graphics depicting an example in which the RL agent would be trained to play Super Mario Bros: https://ibb.co/Sst7L3V
>
> Formally, the theoretical relationship between $I(Z_A;L)$ and the generalisation error is stated in Theorem 3.1: reducing $I(Z_A;L)$ will minimise an upper bound on the generalisation error. We will be sure to make this connection clearer in the final version.
>
> > Line 158: The purpose of Theorem 3.1 is not explained. Also, is it a result from the other paper or proved first time in this paper?
>
> Theorem 3.1 follows directly from Lemma 3.1 in [1]. We will include a sketch proof in the appendix, as the theorem was not explicitly stated in this form in [1].
>
> > Line 165: Why does a high I(Z_C, V) help optimizing the value loss?
>
> A high $I(Z_C; V)$ means that the critic’s learned latent features $Z_C$ are highly informative of the underlying state values $V$. Hence a high $I(Z_C; V)$ helps minimising $\ell_V$.
>
> > Line 166: What do you mean by increasing I(Z_A, V) and what's the relationship between value distillation and increasing I(Z_A, V)?
>
> The standard approach to value distillation, which we employ in this paper, is to learn to predict value targets from $\hat{V}^{aux}=v^{aux} \circ \phi_{A}$, $v^{aux}$ being parameterized as an additional head of the actor network.
>
> Since distillation gradients are propagated through $ \phi_{A}$, performing distillation will increase $I(Z_A; V)$. We point the reviewer to l114-123 for a more detailed description of value distillation.
>
> > Line 175: What do you mean by "whether it is possible to identify latents (ϕ(o), ϕ(o′))) obtained from consecutive observations from pairs stemming from non-consecutive observations?" Is it consecutive or not?
>
> We meant here that a high $I(Z;Z’)$ implies that it is easier for a binary classifier to predict whether a pair of latents $(z_1,z_2)$ corresponds to two consecutive observations $(o, o’)$, or to two non-consecutive observations $(o_x, o_y)$. We will clarify this in the main text.
>
> > Line 194: What's the role of linear probing here?
>
> Linear probing means freezing $\phi^*$ and optimising the learning objective using a linear layer conditioned on $\phi^*(o)$. We will rework the definition of $\phi^*$ in the next revision as we have come to the conclusion that mentioning linear probing adds unnecessary complexity.
>
> > Equation 6: What are $z^*_{A_0}$ and  $z^*_{A_1}$?
>
> $z^*_{A_0}$ and $z^*_{A_1}$ are the two possible outputs of $\phi^*_A$ for the assembly line environment. Can the reviewer clarify what didn't seem clear here?
>
> > Why I((Z^*_A, Z^*'_A); A) is guaranteed to be maximized when following the optimal policy?
>
> Our definition for $\phi^*_A$ in Equation (6) states that we can always identify the optimal action from the value taken by $z^*_A$. It directly follows that $I(Z^*_A;A)$ is always maximised, and, by extension, $I((Z^*_A;Z^{*'}_A);A)$ is maximised as well.
>
> [1] Garcin, S., Doran, J., Guo, S., Lucas, C. G., & Albrecht, S. V. (2024). DRED: Zero-Shot Transfer in Reinforcement Learning via Data-Regularised Environment Design. In Forty-first International Conference on Machine Learning.

---

> > ### Author Response · Authors · 2024-11-23
> > **Authors' response to review (2/2)**
> >
> > > Line 256: I don't see why zero-shot transfer is evident in your example.
> >
> > We believe the reviewer meant the statement in l258. Assuming $\phi^*_{A}$ maps to the reduced MDP in the bottom left of Figure 2 for all levels, including unseen ones, then learning the policy that maps $z^*_{A_0}$ to $a_0$ and $z^*_{A_1}$ to $a_1$ will transfer zero-shot in all levels.
> >
> > > Figure 3: How should one interpret these values? The higher the better? The lower the better? Better summarize the takeaway in the caption.
> >
> > The main takeaways of Figure 3 are as follow:
> > - In Procgen, the actor and critic specialise in extracting different quantities from the agent's observations once they are decoupled.
> > - This specialisation is mostly consistent with the specialisation we expect for the actor or critic representations, which we cover in Section 4.1.
> >
> > In other words, whether we expect a metric to increase or decrease depends on whether we are considering the actor or the critic representation. We will amend the caption to highlight these takeaways.
> >
> > > Line 374: Why does maximizing J^\pi bias \pi to collect trajectories with high I(O; V)?
> >
> > $\phi_A$ is only optimised through $J_\pi$ for decoupled PPO. A high $I(Z_A;V)$ for decoupled PPO can only be explained by the fact that optimising $J_\pi$ somehow causes $I(Z_A;V)$ to increase. Now there are only two mechanisms that could cause $I(Z_A;V)$ to increase:
> > - $\phi_{A}$ gets better at extracting information about $V$ from $O$, i.e. $C(Z|O;V) increases. This is the first hypothesis, stated in l.370.
> > - The observations collected using $\pi$ contain more information about V, i.e. $I(O;V)$ increases. This is the second hypothesis, stated in l.373.
> >
> > > Line 431: Why does actor objective promote learning invariant quantity?
> >
> > In this paragraph, we point out that certain representation learning objectives promote learning invariances to certain quantities implicitly. The quantity in question may not even appear in the formulation of the loss function. For example, when transition dynamics are similar in all levels (e.g. the "left" action moves the controlled character to the left, etc), then performing dynamics prediction promotes learning representations that are level-invariant (i.e. with minimal $I(Z;L)$).
> >
> > We then hypothesise that promoting certain invariances, even implicitly, may be an important factor in determining whether a representation learning objective is effective or not.
> >
> > We believe that the current manuscript is not stating the above hypothesis clearly. In the upcoming revision, we will put an increased focus on clearly communicating the key takeaways in Section 6.
> >
> > To summarise: We will make revisions to the text of the paper to improve readability and make the answers to these questions more clear.

---

> > ### Comment · Reviewer_qthx · 2024-11-24
> >
> > Thanks for clarifying most of my questions. I would suggest adding that explanation to the paper.
> >
> > > A high $I(Z_C; V)$ means that the critic’s learned latent features  are highly informative of the underlying state values. Hence a high $I(Z_C; V)$ helps:
> >
> > This explanation reads informal. Is that just an intuition?

---

> > > ### Author Response · Authors · 2024-11-28
> > > **Response to reviewer follow-up question**
> > >
> > > We thank the reviewer for increasing their score. We understood the reviewer's follow-up question as asking if there was a more formal justification regarding the relationship between $I(Z;V)$ and $\ell_V$. We provide a more formal argument below, which follows a reasoning similar to the argument presented in [1] (Section 3).
> > >
> > > For a batch of $n$ latent $Z$ and value targets $V$, we can write $\ell_V$ as
> > >
> > > $\ell_V (\xi ; Z, V) = \frac{1}{n} \sum_i^n (\tilde{v}^{(i)} - v^{(i)})^2 ,$
> > >
> > > where $\tilde{v}^{(i)} = \hat{v}_\xi (z^{(i)})$ is the value estimate obtained from $z$.
> > >
> > > If we assume residuals roughly take a normal distribution when $\xi \rightarrow \xi^*$, i.e. $ v \approx \tilde{v} + \epsilon, \epsilon \sim \mathcal{N}(0, \sigma) $ , we can approximate $p(v|z)$ as the normal distribution $\mathcal{N} ( \tilde{v}, \sigma )$.
> > >
> > > Since we can express the mutual information as $I(Z;V) = \int dv \ dz \ p(v,z) \log p(v|z) + H(V)$, if we approximate $p(v,z)$ as the empirical distribution $p(v^{(i)}, z^{(i)})$, we have
> > >
> > > $I(Z;V) \approx - \frac{1}{2\sigma^2} \frac{1}{n}  \sum_i^n (\tilde{v}^{(i)} - v^{(i)})^2 -\log(\sigma) -\frac{1}{2}\log(2\pi) + H(V)$
> > >
> > > The first term on the r.h.s. is equivalent to $ - \frac{1}{2\sigma^2} \ell_V (\xi^* ; Z, V)$ and the other terms are constants if $Z$ and $V$ are fixed. Then we see that $I(Z;V)$ and $\ell_V$ would be inversely proportional to one-another, provided that $\xi$ is close to $\xi^*$ .
> > >
> > > [1] Alemi, A. A., Fischer, I., Dillon, J. V., & Murphy, K. (2017). Deep variational information bottleneck. ICLR.

---

### Official Review · Reviewer_fkit · 2024-11-06

**Soundness:** 3
**Presentation:** 3
**Contribution:** 3
**Rating:** 6
**Confidence:** 2

**Summary:**

This paper focus on the difference in representation graph between actor and critic in AC methods in RL. The authors aims to more formally validate an empirical finding where decoupling actor and critics may help with representation learning through

**Strengths:**

- The way in which the topic is approached in the section is well considered, I find it readable
- I appreciate the clear problem formulation and motivation of the paper
- The results, while not counter-intuitive nor surprising, gives some form of closure to the decoupling findings in earlier works, and brings this issue to attention does help clarify the topic
- Assumption seems reasonable

**Weaknesses:**

- Potentially need more ground work setup, such as in the problem definition where an observation function is present, which normally implies that the MI between state and observation is not maximized, but it seems the rest of the paper operate on the assumption that it is indeed maximized and so that the environments are fully observable.
- Evaluation is lacking in breath, it would be great to include a general set of tasks.

**Questions:**

- Why use the idea of context, if I'm not mistaken, the entire work can be done with a normal MDP and MDPs with contexts can be seen as a encompassing MDP with different starting states.
- The increased compute and parameter size of decoupled approach may also play a role in learning, is there any intuition or validation on the topic?
- I disagree that "an optimal or near-optimal representation can never be reached under a shared architecture"; from the reduced MDP point of view, I agree that a shared architecture may not be as concise, but still can be optimal (for both policy and value) given that it is a shared architecture.

---

> ### Author Response · Authors · 2024-11-23
> **Authors' response to review**
>
> We thank the reviewer for their insightful review. We will address the reviewer’s comments below.
>
> > it seems the rest of the paper operate on the assumption that it is indeed maximized
>
> This was not our intention. The theoretical and experimental results presented In this work cover the more general setting in which an observation may not provide all the information about the underlying state. The toy example presented in Figure 2 depicts a fully observable environment, but this was for convenience and to make the environment easy to understand.
>
> If there are other parts of the manuscript that suggest we are assuming full observability, could the reviewer point to them for us to make amendments? We are happy to revise this text to ensure our setting is clear.
>
> > Evaluation is lacking in breath, it would be great to include a general set of tasks
>
> The new revision will include experiments conducted in a continuous control benchmark in several Brax environments with video distractors, similar to [1]. We point the reviewer to the [proposed changes to manuscript] comment for details on our planned experiments.
>
> > Why use the idea of context, if I'm not mistaken, the entire work can be done with a normal MDP and MDPs with contexts can be seen as a encompassing MDP with different starting states.
>
> As pointed out by the reviewer, it is possible to reframe any CMDP as a (partially observable) MDP. However, by explicitly separating the environment properties that must remain constant over an episode (i.e. the context), the CMDP provides several advantages:
> - It allows splitting environment instances (levels) into training & test sets. This minimises leakage between the train and test distributions, leading to a more rigorous experimental setup where we can enhance our study of generalisation.
> - Explicitly defining the context lets us derive theoretical insights about the overfitting behaviour of RL agents (see for example theorem 3.1). These insights can be leveraged into metrics (e.g. $I(Z;L)$) to quantify how much the agent’s representation overfits to its training data.
>
> We point to [2] for additional analysis and discussion on the advantages of employing the CMDP framing.
>
> > The increased compute and parameter size of decoupled approach may also play a role in learning, is there any intuition or validation on the topic
>
> We are currently conducting additional experiments in which we match the number of parameters between the coupled and decoupled architectures. We will include our results in the next revision.
>
> > I disagree that "an optimal or near-optimal representation can never be reached under a shared architecture"; from the reduced MDP point of view, I agree that a shared architecture may not be as concise, but still can be optimal (for both policy and value) given that it is a shared architecture.
>
> We believe this disagreement stems from whether one considers compactness a necessary property for an optimal representation. Currently, the notion that the optimal representation yields the most compact embedding from which the learning objective can be optimised is made implicit by our choice of $Z^*$  in l196. However it was not explicitly stated when we define the optimal representation in l194. We will amend the definition in the next revision to reflect this notion.
>
> [1] Stone, A., Ramirez, O., Konolige, K., & Jonschkowski, R. (2021). The Distracting Control Suite--A Challenging Benchmark for Reinforcement Learning from Pixels. arXiv preprint arXiv:2101.02722.
>
> [2] Kirk, R., Zhang, A., Grefenstette, E., & Rocktäschel, T. (2023). A survey of zero-shot generalisation in deep reinforcement learning. Journal of Artificial Intelligence Research, 76, 201-264.

---

### Official Review · Reviewer_Cooi · 2024-11-09

**Soundness:** 3
**Presentation:** 2
**Contribution:** 4
**Rating:** 6
**Confidence:** 3

**Summary:**

The paper investigates representation learning objectives for actor and critic networks in deep reinforcement learning, specifically in the context of policy gradient algorithms. Using information-theoretic metrics, the study reveals that the actor and critic networks naturally specialize in capturing distinct information types, with the actor focusing on action-relevant features and the critic on value and environmental dynamics. This specialization is shown to enhance model architecture design and improve representation learning objectives, ultimately contributing to greater sample efficiency in reinforcement learning. In addition, this paper also how various representation learning objectives could help shaping the actor and critic's representation and improve the RL sample efficiency.

**Strengths:**

-  **Novel Perspective on Representation Specialization**: This paper provides a fresh and thorough analysis of actor and critic representation specialization using information-theoretic metrics.
- **Convincing Empirical Evidence**: The authors provide substantial empirical support for their claims, with extensive experiments that effectively demonstrate the impact of decoupling actor and critic representations, as well as impacts of different representation learning objectives on the resulting representation.
- **Insightful Conclusions**: The study draws interesting and impactful conclusions regarding how specialized representation learning objectives can improve sample efficiency and generalization in RL, a great contribution to the RL community, shedding light on the design of future deep RL algorithms.

**Weaknesses:**

- **Limited Scope of Algorithm Selection**: The paper exclusively focuses on online, on-policy, policy gradient algorithms (e.g., PPO, DCPG), leaving it unclear how well the conclusions apply to other RL frameworks, particularly off-policy algorithms like DDPG or SAC. Broader consideration of diverse algorithm types would strengthen the generalizability of the findings. Also, whether the same conclusion holds under different model sizes? It would be useful to add in the paper to show that this conclusion would still hold if we simply added capacity to the actor/critic network without changing any learning objectives.
- **Presentation and Clarity**: The paper’s structure and presentation could be significantly improved to make it more easier to understand for readers. The writing is currently dense, making it challenging to grasp the main takeaways and hypotheses. For instance, restructuring sections with clearer headings, especially in Sections 6.1 and 6.2, would help readers navigate the study’s conclusions. Using bullet points to highlight key hypotheses and findings, as well as starting each paragraph with a bolded summary sentence, could enhance readability and ensure that readers capture the core messages more effectively.

**Questions:**

- What does the delta percentage in Figure 1 represent? It would be helpful if the authors added a brief explanation to the caption.
- How is mutual information measured in this study? Currently, it’s detailed only in the appendix, but a high-level explanation in the background section would help readers understand the methodology better.
- Could the authors add a performance plot to the figures in the main paper? This would clarify how the information-theoretic metrics relate to final performance outcomes.
- Is it true that $I((Z,Z'); A)$ + $I(Z,Z')$ in general is a good indicator of the algorithm's performance? (Looks like it's the case from Figure 3 and Figure 7.) The authors seem to argue that while actor representation (decoupled) tends to have larger $I((Z,Z'); A)$ while smaller $I(Z,Z')$ and the reverse holds for critic representation, combing them in a coupled representation does not combine the best of both world and resulting in smaller $I((Z,Z'); A)$ + $I(Z,Z')$ and thus worse performance?
- Could Figures 7 and 8 be moved from the appendix into the main paper? The authors might need to shorten other parts of the text rather than placing key results in the appendix while referring back to them in the main sections.
- In Section 6.1, it would be clearer if the authors defined the dynamics prediction objective from Moon et al. with a brief equation or sentence rather than just citing the source. Similarly, in Section 6.2, could the authors introduce the MICo objective with an equation first?
- In Section 6.1, lines 466–468, the distinction between explanations (1) and (2) is unclear; both seem to attribute reduced overfitting to the larger batch size used in value distillation. If the authors hypothesize that batch size is the main factor here, could they test this by running PPO with larger batch sizes, then measuring the relevant metrics to assess any changes in overfitting?
- A broader concern is how the study's conclusions about decoupled versus coupled objectives hold across varying model sizes. For instance, in Section 6.2, the authors note that adding a dynamic prediction objective to the critic impacts certain metrics but harms performance due to an unintended specialization. Similarly, the study attributes overfitting to the coupled representation. Could these conclusions change with larger model capacity (and also training with larger batch sizes as the author claims that this results in more overfitting)? Intuitively, if the shared representation had sufficient capacity (and were trained on larger batches as discussed), could a shared representation still achieve strong performance across the information-theoretic metrics, potentially improving performance without conflicting specializations?

---

> ### Author Response · Authors · 2024-11-23
> **authors' response to review**
>
> We thank the reviewer for their thorough and insightful review. We agree with the reviewer that the clarity and presentation of the manuscript can be improved, and will incorporate changes to the manuscript in light of their suggestions. We point the reviewer to the comment titled [proposed changes to manuscript] at the top for a summary of the changes we plan to make for the next revision. We will use this thread to address the reviewer’s questions.
>
> > What does the delta percentage in Figure 1 represent? It would be helpful if the authors added a brief explanation to the caption.
>
> We have reworked this table (Figure 1, right) as a standalone table with more information. See this link for a preview of the new table that will be included in the next revision: https://ibb.co/vwHnfxS
>
> > How is mutual information measured in this study? Currently, it’s detailed only in the appendix, but a high-level explanation in the background section would help readers understand the methodology better.
>
> > Could the authors add a performance plot to the figures in the main paper? This would clarify how the information-theoretic metrics relate to final performance outcomes.
>
> > Could Figures 7 and 8 be moved from the appendix into the main paper? The authors might need to shorten other parts of the text rather than placing key results in the appendix while referring back to them in the main sections.
>
> Restructuring section 5 and 6 (see [proposed changes to manuscript] comment) will free up space in the main text and allow us to include this information. Namely:
> - How mutual information is measured (currently in appendix)
> - Train and test scores achieved by the agents (currently in appendix)
>
> > In Section 6.1, it would be clearer if the authors defined the dynamics prediction objective from Moon et al. with a brief equation or sentence rather than just citing the source. Similarly, in Section 6.2, could the authors introduce the MICo objective with an equation first?
>
> We will provide a detailed description of each representation learning objective tested in the new revision. Whether this information is included within the main text or within the appendix will be contingent on the available space.
>
> > Is it true that $I((Z,Z'); A) + I(Z,Z')$ in general is a good indicator of the algorithm's performance? (Looks like it's the case from Figure 3 and Figure 7.) The authors seem to argue that while actor representation (decoupled) tends to have larger $I((Z,Z'); A)$ while smaller $I(Z,Z')$ and the reverse holds for critic representation, combing them in a coupled representation does not combine the best of both world and resulting in smaller $I((Z,Z'); A) + I(Z,Z')$ and thus worse performance?
>
> We do not observe a clear connection between $I(Z,Z’;A) + I(Z;Z)$ and agent performance. For example:
> - DCPG[sh] significantly outperforms PPO[sh], but they have similar $I(Z,Z’;A) + I(Z;Z)$.
> - DCPG outperforms PPO, but they have similar $I(Z,Z’;A) + I(Z;Z)$ for both the actor (unhashed bar in figure 3) and critic (hashed bar in figure 3).
> - PPG and DCPG perform relatively similarly, but PPG has greater $I(Z,Z’;A) + I(Z;Z)$ than DCPG for both the actor and critic.
>
> > In Section 6.1, lines 466–468, the distinction between explanations (1) and (2) is unclear; both seem to attribute reduced overfitting to the larger batch size used in value distillation. If the authors hypothesize that batch size is the main factor here, could they test this by running PPO with larger batch sizes, then measuring the relevant metrics to assess any changes in overfitting?
>
> > A broader concern is how the study's conclusions about decoupled versus coupled objectives hold across varying model sizes. For instance, in Section 6.2, the authors note that adding a dynamic prediction objective to the critic impacts certain metrics but harms performance due to an unintended specialization. Similarly, the study attributes overfitting to the coupled representation. Could these conclusions change with larger model capacity (and also training with larger batch sizes as the author claims that this results in more overfitting)? Intuitively, if the shared representation had sufficient capacity (and were trained on larger batches as discussed), could a shared representation still achieve strong performance across the information-theoretic metrics, potentially improving performance without conflicting specializations?
>
> We thank the reviewer for suggesting these experiments, we see how they could help substantiate some of the hypotheses we made in Section 6. We point the reviewer to the [proposed changes to manuscript] comment for details on our planned experiments.

---

> > ### Comment · Reviewer_Cooi · 2024-11-28
> >
> > Thanks the authors for the rebuttal. I wonder why the updated manuscript is not uploaded to OpenReview directly? In this way, we could see the diff between the original manuscript and the updated one. Also I have trouble opening up the link of the revised manuscript.

---

> > > ### Author Response · Authors · 2024-11-28
> > > **Response to reviewer follow-up question**
> > >
> > > Please see the updated PDF. You should now be able to see the revised manuscript. Please do let us know if you have issues opening the new version.
> > >
> > > We again thank the reviewer for their time and useful feedback.

---

### Author Response · Authors · 2024-11-23
**Proposed changes to manuscript - part 1: presentation and clarity improvements**

We thank all of the reviewers for their feedback and for spending their time on suggestions that would improve the current manuscript. We will outline here the improvements we have made and plan to make for the new version of the manuscript. We believe these changes address the vast majority of the reviewers' comments and concerns. Following our summary of the reviewers' observations and our responses, we reply directly to each reviewer’s specific questions and comments.

**PRESENTATION & CLARITY IMPROVEMENTS**

Several reviewers noted that the presentation and clarity of the paper can be improved. We agree with their assessment. We will list the main issues we have identified from the reviews, and how we are planning to address each.

**Issue n1 - Key takeaways from the study should be stated more clearly.** The reviewers pointed out that it can be challenging to identify the key takeaways in certain sections, and in particular in Section 6.

Broadly, there are four key takeaways from this study:
- The actor and the critic have different optimal representations (covered in Section 4.1)
- Once decoupled, the actor and critic specialise in extracting different information from observations. In practice, their specialisation is broadly consistent with what we expect from their respective optimal representation. (covered in Section 4.2)
- For the actor, representation learning objectives are effective when they promote capturing information shared across all training levels, either explicitly or implicitly. (covered in Section 6.1)
- The critic is capable of influencing exploration to benefit its own learning objective (covered in Section 5), and auxiliary objective(s) (covered in Section 6.2).

To ensure the above takeaways are clearly communicated, we will:
- Rework the structure of sections 5 and 6 so that each section focuses on a single takeaway.
- Ensure these takeaways get prominently featured within their respective section, but also in the introduction, abstract and conclusion.
- Rework certain figures (for example figure 1) to improve how they communicate this information.


**Issue n2 - It is difficult to keep track of what each of the four mutual information (MI) metrics measures and what they say about the representation.** We believe this problem stems from the reader not being introduced to the different MI metrics until Section 3. Furthermore, the section is long and contains a lot of mathematical notation, making it difficult to refer back to.

To address this issue, we will rework the table in Figure 1 (right) as a standalone Table containing compact and informal definitions for each MI metrics and what they quantify. The new version can be seen at https://ibb.co/vwHnfxS

In doing so, we aim to familiarise the reader with the MI metrics as early as the introduction and to have the new Table 1 be a "cheatsheet" to refer to when going through the rest of the paper.
In addition, we will make a range of minor improvements to improve clarity further, outlined in individual comments to the reviewers.

---

> ### Author Response · Authors · 2024-11-23
> **Proposed changes to manuscript - part 2: additional experiments**
>
> **ADDITIONAL EXPERIMENTS**
>
> We thank the reviewers for suggesting several additional experiments. We are currently running the following experiments. We will include results in the next revision.
>
> **Additional experiments in a continuous control domain.** The new revision will feature additional experiments conducted in a continuous control benchmark formed of several Brax environments with video distractors, similar to [1]. The new experiments match the scope of our Procgen experiments. They confirm that the key insights we identified in the original study also apply in a different domain.
>
> **Additional representation learning objective.** We add Data-Regularised Actor Critic (DrAC) [2] as an additional representation learning objective baseline. DrAC explicitly encourages learning invariances to different augmentations of the same observation. The objectives we had tested so far focused on adding new information (i.e. through prediction tasks) and the removal of superfluous information (i.e. regularisation) occurred implicitly. In contrast, DrAC performs regularisation explicitly, and therefore it fills a gap in our study.
>
> **Investigation on the effect of scaling the number of parameters.** Following suggestions made by the reviewers, we are conducting experiments with scaled-up architectures of PPO[sh] and PPO (decoupled). We aim to answer the following questions:
> - Do we observe the same actor and critic specialisation as representation capacity increases?
> - Is additional information being extracted?
> - In our initial experiments, PPO (decoupled), fits ~2x as many parameters as PPO[sh], since it has separate representations. How does PPO[sh] compare to decoupled PPO at equal parameter counts?
>
> **Investigation on the importance of the batch size & data diversity for representation learning.** In Section 6.1, we claim that value distillation improves performance in PPG, but not in PPO, because the batch of data used each time PPG performs value distillation (i.e. for the auxiliary phase) is 32 times larger. We hypothesise that a large batch size is a necessary condition to promote encoding information shared across training levels over information specific to individual levels, and improve performance for PPG. To substantiate this claim, we follow the approach taken in [3]. We are conducting an experiment where we gradually reduce the auxiliary phase batch size in PPG from 32x PPO down to 1x PPO's, while keeping the total number of distillation updates constant.
>
> Lastly, we note that multiple reviewers have suggested conducting a similar analysis for off-policy algorithms. We concur with the reviewers that this analysis would represent a valuable contribution to the field. However, we have decided to leave this study to future work. We do so for two reasons:
> 1) We currently do not have access to the additional compute that would be required for conducting this study.
> 2) This study would likely highlight nuances and complexities that are unique to the off-policy setting. We believe this would make the scope of this work too broad for a conference paper. The 10 page limit would be ill-suited for imparting the additional information to the reader, and it would adversely affect the clarity of the current paper contents.
>
> We are working towards a new version of the manuscript, which we will upload before the end of the discussion period. Whenever possible, we will post here partial results for any additional experiment that did not finish by the time we upload the new revision.
>
> [1] Stone, A., Ramirez, O., Konolige, K., & Jonschkowski, R. (2021). The Distracting Control Suite--A Challenging Benchmark for Reinforcement Learning from Pixels. arXiv preprint arXiv:2101.02722.
>
> [2] Raileanu, R., Goldstein, M., Yarats, D., Kostrikov, I., & Fergus, R. (2021). Automatic data augmentation for generalization in reinforcement learning. Advances in Neural Information Processing Systems, 34, 5402-5415.
>
> [3] Wang, K., Zhou, D., Feng, J., & Mannor, S. (2023). PPG reloaded: an empirical study on what matters in phasic policy gradient. Proceedings of the 40th International Conference on Machine Learning (ICML'23), Vol. 202. JMLR.org, Article 1526, 36694–36713.

---

### Author Response · Authors · 2024-11-28
**New and improved version of the manuscript**

We thank the reviewers again for their helpful feedback which improved the paper significantly. Please see the new version of the manuscript, which includes the reviewers' suggestions for experimentation with the number of learned parameters, experimentation in continuous control environments, and experimentation with batch sizes. We have also included more experiments with additional auxiliary objectives and base algorithm combinations. Due to the large number of algorithms studied in this work, many plots are half the length of an entire page and, therefore, must be delegated to the appendix. However, we ensured that the paper's critical points are supported by plots in the main body of the paper by extracting specific elements from the full plots in the appendix.

Finally, we have improved the paper's writing and presentation clarity. For example, we now include a table at the beginning of the paper that clearly defines each of the mutual information measures we consider and gives an example of what each quantity captures.

If selected for publication, we plan to further improve the paper by including even more detailed information in the appendix, such as the training curves for each algorithm in each of the environments tested, detailed descriptions (in addition to the citations) for each representation learning objective we study, and more.

---

### Meta-Review · Area_Chair_8EAa · 2024-12-20

**Metareview:**

This paper studies the representation of actor and critic in on-policy RL algorithms using information theory. Overall, the reviewers are positive about the paper. Multiple reviewers raises concerns on clarity during the discussion which I think is mostly addressed by the authors' response. Some reviewer mention that it's unclear how the current paper's finding may apply to off-policy algorithms. I think this is a serious concern. The author should clearly highlight the scope of algorithms that are analyzed here e.g. in title, abstract, and introduction (e.g. main finding summaries). The current writing (e.g. title and abstract) suggests a very broad over claim for all actor critic algorithms, which can be misleading.

**Please fix these writing issues and clarify the scope the findings are this paper is applicable to.**

**Additional Comments On Reviewer Discussion:**

Reviewer Cooi, Reviewer fkit, Reviewer qthx  Reviewer 5hR mention concern on writing clarity, Reviewer Cooi and Reviewer 5hRY  mention limited scope of algorithms the finding here can apply (it's unclear whether off-policy algorithms have the same property). I found that the authors never responded to this issue of scope in the discussion, which however is a real concern. It's important to clarify and highlight the limitation of this work.

---

### Decision · Program_Chairs · 2025-01-22

Accept (Poster)